# DISCRETIZATION-INVARIANCE? ON THE DISCRETIZATION MISMATCH ERRORS IN NEURAL OPERATORS

**Wenhan Gao**[1]    **Ruichen Xu**[1]    **Yuefan Deng**[1]    **Yi Liu**[1,2*]
[1] Department of Applied Mathematics and Statistics, Stony Brook University
[2] Department of Computer Science, Stony Brook University

## ABSTRACT

In recent years, neural operators have emerged as a prominent approach for learning mappings between function spaces, such as the solution operators of parametric PDEs. A notable example is the Fourier Neural Operator (FNO), which models the integral kernel as a convolution operator and uses the Convolution Theorem to learn the kernel directly in the frequency domain. The parameters are decoupled from the resolution of the data, allowing the FNO to take inputs of different resolutions. However, training at a lower resolution and inferring at a finer resolution does not guarantee consistent performance, nor can fine details, present only in fine-scale data, be learned solely from coarse data. In this work, we address this misconception by defining and examining the discretization mismatch error: the discrepancy between the outputs of the neural operator when using different discretizations of the input data. We demonstrate that neural operators may suffer from discretization mismatch errors that hinder their effectiveness when inferred on data with resolutions different from that of the training data or when trained on data with varying resolutions. As neural operators underpin many critical cross-resolution scientific tasks, such as climate modeling and fluid dynamics, understanding discretization mismatch errors is essential. Based on our findings, we propose a Cross-Resolution Operator-learning Pipeline that is free of aliasing and discretization mismatch errors, enabling efficient cross-resolution and multi-spatial-scale learning, and resulting in superior performance. The code is publicly available at `https://github.com/wenhangao21/ICLR25-CROP`.

## 1 INTRODUCTION

In diverse fields across science and engineering, researchers aim to investigate the behavior of physical systems under different parameters, such as different initial conditions or forcing functions. Traditional numerical approaches often prove to be excessively time-consuming for simulating parametric physical systems. Data-driven surrogate models, known as neural operators (NOs), present an efficient alternative (Anandkumar et al., 2019; Raonic et al., 2023; Lu et al., 2021). Neural operators learn the mapping from parameter function space to the solution function space. After the training phase, performing inference only necessitates a forward pass of the network, which can be several orders of magnitude faster than traditional numerical methods. A particular example is the Fourier Neural Operator (FNO) (Li et al., 2021), which models the integral kernel as a convolution operator and employs the Convolution Theorem to learn the kernel directly in the frequency domain to decouple the parameters from the resolution of the data.

However, there is a long-standing misconception in the operator learning community: It is widely believed that training an FNO on one resolution allows inference on another without degrading its performance, since FNO operates and parameterizes the kernel in the Fourier space. In reality, it still exhibits a strong bias toward the training resolution. We attribute this inconsistency across different resolutions to discretization error. Traditionally, in numerical methods, discretization error refers to the discrepancy between continuous mathematical models and their discrete approximations, which arises due to the finite resolution of computational systems. In the context of neural operators, the continuous mathematical model is represented when the neural operator $\mathcal{G}(\cdot;\theta)$, parameterized by network parameters $\theta$, takes a continuous function $a$ as input: $\mathcal{G}(a;\theta)$. In contrast, the discrete

---

*Correspondence to yi.liu.4@stonybrook.edu

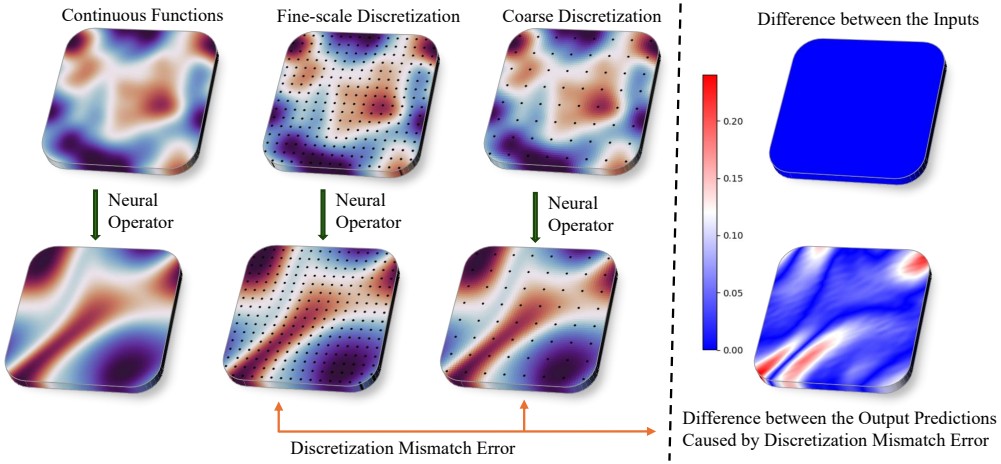

Figure 1: Neural operators map between function spaces and can handle different discretizations. However, their architectures still introduce discretization errors. In computational math, researchers analyze the discrepancies between continuous processes and their discrete approximations. ***In the case of FNOs, we never have the continuous model that is trained with continuous data***. Instead, we define discrepancies between two discrete approximations as discretization mismatch errors (DMEs) in Sec. 4.1. Minimizing DMEs is crucial in cross-resolution tasks.

approximations occur when the neural operator takes a discretization of the continuous function as input: $\hat{\mathcal{G}}(a|_{\Omega_J}; \theta)$, where $a|_{\Omega_J}$ denotes a discretization of $J$ observation points of the function $a$. In practice, network parameters are optimized on discretizations of functions; therefore, we argue that for neural operators, it is essential to minimize the discrepancy between different discrete representations. We will refer to this type of error as the discretization mismatch error (DME).

DMEs are caused by the network architectures; a simple example is Convolutional Neural Networks (CNNs), which are generally not considered discretization-invariant models. When the resolution of the input data converges to infinity, discrete convolution, whose kernel remains fixed in size, converges to a point-wise operation (Liu-Schiaffini et al., 2024). In this work, we analyze the DME in FNO, which is the most popular discretization-invariant neural operator. This work is a prototypical analysis of DMEs in neural operators, especially for mesh-based neural operators. We refer to tasks where the inference resolutions differ from the training resolution as cross-resolution tasks for the reasons given in Sec. 4.1, and the analysis of DMEs is pivotal for such tasks. As an integral neural operator, the mathematical formulation of FNO operates in the continuum, representing a groundbreaking approach to function space mappings via neural networks. However, the realization of FNO is still at a discrete level as the inputs and intermediate outputs are discrete; thus, we argue that FNO suffers from DMEs. Our work does not seek to criticize FNO in any way but rather builds upon this revolutionary framework while highlighting the issue of DMEs for the community's consideration to advance the designs of neural operators.

In this work, we study and locate the sources of DMEs. Both theoretical analysis and empirical evaluation are provided to convey our findings. Moreover, neural operators may suffer from aliasing errors that stem from inadequate sampling of high-frequency contents introduced by the neural operator (Raonic et al., 2023). To mitigate these issues, we propose the Cross-Resolution Operator-learning Pipeline (CROP); CROP is free of both DMEs and aliasing errors. We summarize and compare several popular neural operators in Table 1. **Our contributions can be summarized in four parts:** (1) We address this important, yet often overlooked, issue of the accumulation of DMEs in neural operators. (2) We provide mathematical analysis and empirical evaluation to identify the sources and accumulation of DMEs. (3) We design CROP components that capture global patterns and are robust in cross-resolution tasks. (4) CROP represents a general pipeline in which the intermediate neural operator can be freely chosen to map the feature functions between band-limited function spaces, such as CNN-based networks for fine-grained feature extraction.

## 2 RELATED WORK

Over the past few years, neural PDE solvers have emerged as a promising alternative to conventional numerical methods for solving PDE problems across various domains in practical engineering and

Table 1: A comparison of various popular architectures for learning function space mappings.

| Architecture | Discretization Invariance | Free of Aliasing Error | Free of DME | Efficiency |
|---|---|---|---|---|
| FNO | ✓ | ✗ | ✗ | ✓ |
| U-Net | ✗ | ✗ | ✗ | ✓ |
| CNO | ✓* | ✓* | ✓ | ✗* |
| DeepONet | ✓! | ✓ | ✗! | ✓ |
| CROP (ours) | ✓ | ✓ | ✓ | ✓ |

In Sec. 4.1, the notions of discretization invariance and discretization mismatch error are introduced.

The notion of aliasing error (continuous-discrete equivalence) is introduced in Raonic et al. (2023).

* Interpolation is employed for resolution independence and being free of discretization mismatch errors.

* Interpolation operations in up and down samplings for activation functions can be computationally costly.

! DeepONet is only resolution independent on the output.

life sciences (Sirignano et al., 2020; Pathak et al., 2022; Zhang et al., 2022; Azizzadenesheli et al., 2024; Zhang et al., 2023). Traditionally, solving a PDE involves seeking a smooth function satisfying the derivative constraints imposed by the equations. From this perspective, Physics-Informed Neural Networks (PINNs) (Raissi et al., 2019) have been developed to approximate solutions of PDEs individually. An alternative perspective views differential operators as mappings between function spaces, where the solution operators are the inverses of these mappings. Building upon this viewpoint, neural operators have been introduced as a method for solving a family of parametric PDEs (Anandkumar et al., 2019; Lu et al., 2021; Brandstetter et al., 2022; Fanaskov & Oseledets, 2024; Raonic et al., 2023). Neural operators parameterize solution operators with neural networks to map the input parameter functions to their respective solutions directly. More generally, beyond the solution operators of PDEs, neural operators can approximate other function-to-function mappings, including derivative operators, integral operators, etc..

Integral neural operator is a general framework that parameterizes the operator as a sequence of kernel integrals and nonlinear activations (Kovachki et al., 2023; Anandkumar et al., 2019). FNO is a type of integral neural operator in which the kernel integral is imposed to be a convolution (Li et al., 2021). Global convolution is performed via Fourier layers in the frequency domain. By leveraging low-frequency modes and truncating high-frequency modes, FNO captures global information while incurring low computational costs and achieving discretization invariance. Although FNO is discretization-invariant, its inconsistent performance in super-resolution tasks has been noted in several works (Raonic et al., 2023; Li et al., 2022). Mesh-based neural operators including FNO, GNO (Anandkumar et al., 2019), CNO (Raonic et al., 2023), and DeepONet (Lu et al., 2021) either are not completely discretization-invariant or suffer from DMEs. While function-basis models, such as SNO (Fanaskov & Oseledets, 2022), may not suffer from DMEs, they might be inferior in capturing spatial information and dependencies. In this work, we specifically analyze the DMEs in FNO, a model that is not intuitively expected to encounter such issues.

**Relations with Prior Work.** In Li et al. (2022), the inconsistent super-resolution performance has been noted and the authors propose to mitigate this issue by using physics-informed constraints to fine-tune the neural operator. However, this method is limiting, as different physics-informed constraints (losses) can be difficult to optimize and balance (McClenny & Braga-Neto, 2020; Gao & Wang, 2023; Wang et al., 2023), and in practical applications, we might have only data available, not the governing PDEs. In Raonic et al. (2023); Bartolucci et al. (2023), aliasing errors in neural operators have been investigated. The notion of continuous-discrete equivalence (CDE) is formalized to study whether there is an equivalence between their continuous and discrete representations in neural operators, i.e., whether the outputs of the neural operator reside in the band-limited function space of interest. Based on CDE, Convolutional Neural Operators are proposed to be an operator adaptation of the widely used U-Net architecture. Their views on aliasing errors in neural operators are constructive and critical. However, while aliasing error is a very important perspective on the design of neural operators, we note that anti-alias activation functions do not overcome the aforementioned issues; this will be discussed further in Sec. 5.1. We provide a different explanation as to why FNO is not robust across different resolutions. In this work, DMEs are defined and analyzed; we propose the Cross-Resolution Operator-learning Pipeline that minimizes DMEs to achieve robust performance across different resolutions. Our CROP components can be seamlessly used in FNO or CNN-based architectures such as CNO to enable efficient learning of both global features and local details while decoupling the network parameters from discretizations of the input data.

## 3 BACKGROUND

### 3.1 OPERATOR LEARNING

Operator learning refers to the use of neural networks to approximate or learn an operator, $G$, mapping between function spaces (Kovachki et al., 2023). Neural operators are particularly relevant in the field of scientific computing, where complex mathematical operators are often involved in solving partial differential equations (PDEs). A particular and commonly seen example of operators is for the solution operator of parametric PDEs. Consider a parametric PDE of the form:

$$\mathcal{N}(a, u)(x) = f(x), \quad x \in \Omega$$
$$u(x) = 0, \quad x \in \partial\Omega$$

where $\Omega \subset \mathbb{R}^d$ is a bounded open set, $\mathcal{N}$ is a, possibly non-linear, differential operator, $a$ is the parametric input function, $f$ is a given fixed function in an appropriate function space determined by the structure of $\mathcal{N}$, and $u$ is the PDE solution. The PDE solution operator is defined as $G(a) = u$.

In practice, for appropriate function spaces $\mathcal{A}$ and $\mathcal{U}$, we are interested in learning an operator $G : \mathcal{A} \mapsto \mathcal{U}$ with a neural operator $\mathcal{G}(\cdot; \theta)$ through a finite collection of observations of input-output pairs $\{a_j, u_j\}_{j=1}^{N_{\text{data}}}$, where $\theta \in \Theta$ is a set of network parameters and $a_j \sim \mu$ are i.i.d. samples drawn from some probability measure $\mu$ supported on $\mathcal{A}$. We aim to control the $L_\mu^2(\mathcal{A}; \mathcal{U})$ Bochner norm of the approximation on average with respect to $\mu$:

$$\min_{\theta \in \Theta} \|G(\cdot) - \mathcal{G}(\cdot; \theta)\|_{L_\mu^2(\mathcal{A}; \mathcal{U})}^2 = \mathbb{E}_{a \sim \mu} \|G(a) - \mathcal{G}(a; \theta)\|_{\mathcal{U}}^2$$

$$= \int_{\mathcal{A}} \|G(a) - \mathcal{G}(a; \theta)\|_{\mathcal{U}}^2 d\mu(a)$$

$$\approx \frac{1}{N_{\text{data}}} \sum_{j=1}^{N_{\text{data}}} \|u_j - \mathcal{G}_\theta(a_j)\|_{\mathcal{U}}^2.$$

Practically, each $a_j$ and $u_j$ are functions that come with a discretization (function values at some sensor points), one could seek to approximately solve the empirical-risk minimization problem directly at those sensor points.

### 3.2 FOURIER NEURAL OPERATORS

Inspired by the kernel method for PDEs, each integral neural operator layer consists of a fixed non-linearity and a kernel integral operator $\mathcal{K}$ modeled by network parameters, defined as $(\mathcal{K}v)(x) = \int \kappa(x, y)v(y)\mathrm{d}y$. All operations in integral neural operators are defined on functions; thus, integral neural operators are understood as function space architectures.

As a natural choice inspired by CNNs and the perspective of fundamental solutions, FNO imposes the integral kernel to be translation invariant, $\kappa(x, y) = \kappa(x - y)$. Thus, the kernel integral operator becomes a convolution operator, and FNO performs global convolution in the frequency domain. Each Fourier layer in FNO consists of a fixed non-linearity and a convolution operator $\mathcal{K}$ modeled by network parameters:

$$(\mathcal{K}v)(x) = \int_{\mathbb{R}^d} \kappa(x - y)v(y)dy. \tag{1}$$

Convolution can be efficiently carried out as element-wise multiplication in the frequency domain:

$$(\mathcal{K}v)(x) = \mathcal{F}^{-1}(\mathcal{F}\kappa \cdot \mathcal{F}v)(x), \tag{2}$$

where $\mathcal{F}$ and $\mathcal{F}^{-1}$ are the Fourier transform and its inverse, respectively. FNO directly learns $\mathcal{F}\kappa$ in the frequency domain instead of learning the kernel $\kappa$ in physical space. The Fourier transform captures global information effectively and efficiently, leading to superior performance for FNO.

## 4 CROSS-RESOLUTION OPERATOR-LEARNING PIPELINE

In this section, we delve into the following questions: (1) Do neural operators exhibit robustness across different resolutions? (2) If not, what factors contribute to this instability? (3) Based on the analysis, how can we improve the resilience of neural operators across various resolutions and potentially achieve additional benefits? We first define DMEs in Sec. 4.1; we answer the first two questions in Sec. 4.2 and the last question in Sec. 4.3. We will use FNO as an example for the analysis of DMEs; however, the concepts and analysis can be extended to other mesh-based neural operators, such as the Wavelet neural operator (Gupta et al., 2021).

### 4.1 DISCRETIZATION ERROR AND DISCRETIZATION MISMATCH ERROR

For simplicity, we consider $H^s\left(\mathbb{T}^d\right)$ and $H^{s'}\left(\mathbb{T}^d\right)$ as the input-output function spaces, a unit torus $\Omega = \mathbb{T}^d = [0,1)^d$ as the domain, and equidistant (uniform) grids as the form of our discretization. The FNO architecture is usually applied to such discretizations to harness the power of the Fast Fourier Transform (FFT). The operations in FNO are mainly defined on general geometries and discretization. The FNO architecture is not limited to rectangular domains, periodic functions, or uniform grids. Likewise, our analysis is not limited by these constraints; however, the derivation can become considerably more tedious; thus, we adopt this simplification.

**Definition 4.1.** A uniform discrete refinement $(\Omega_J)_{J=1}^{\infty}$ of the domain $\Omega = \mathbb{T}^d$ is a sequence of nested sets of equidistant grids $\Omega_1 \subset \Omega_2 \subset \cdots \subset \Omega$ with $|\Omega_J| = Jd$ for any $J \in \mathbb{N}$ such that, for any $h > 0$, there exists a number $J = \lceil \frac{1}{h} \rceil \in \mathbb{N}$ such that

$$\Omega \subseteq \bigcup_{x \in \Omega_J} \left\{y \in \mathbb{R}^d : \|y - x\|_2 < h \right\}.$$

Any member $\Omega_J$ is called a discretization of the domain $\Omega$. Note that, for an $\mathbb{R}^m$-valued function $a : \Omega \to \mathbb{R}^m$ defined on $\Omega$, we call the pointwise evaluation of the function on $\Omega_J$, $a|_{\Omega_J} = \{a\left(x_j\right) : x_j \in \Omega_J\}$, a discretization of the function $a(x)$. Notice that $a|_{\Omega_J}$ can be viewed as a vector in $\mathbb{R}^{Jd}$. Therefore, an FNO that takes discrete inputs can be viewed as a mapping $\hat{\mathcal{G}} : \mathbb{R}^{Jd} \times \Theta \to \mathcal{U}$, in contrast to its continuous counterpart $\mathcal{G} : \mathcal{A} \times \Theta \to \mathcal{U}$. Here, $\hat{\mathcal{G}}$ indicates a pseudo-FNO that handles discrete data inputs. For simplicity, we will omit the $\hat{\cdot}$ notation as the context will make it clear.

As we assume uniform grids, storing the sensor point coordinates, $x_j$, is unnecessary; thus, $a|_{\Omega_J}$ can be directly used as input to the Fourier neural operator. While including coordinates as input to the neural network may sometimes improve performance, as noted in Kovachki et al. (2023). Nevertheless, our analysis still applies if we consider coordinate inputs as the identity function (the function's output at point $x_j$ is $x_j$ itself). For the purposes of discussion in this work, we will treat $a|_{\Omega_J}$ as a vector in $\mathbb{R}^{Jd}$. For a discretization of size $N$, we will view it as a function reconstructed by the $N$ Fourier coefficients. This view is natural from the perspective of the orthogonal Fourier projection operators.

It is well known in the field of operator learning that FNO is regarded as a discretization-invariant model. We herein present the definition of discretization invariance as given in Kovachki et al. (2023), with slight modifications related to our notations and conventions.

**Definition 4.2** (Kovachki et al. (2023))**.** Let $\Theta \subseteq \mathbb{R}^p$ be a finite-dimensional parameter space and $\mathcal{G} : \mathcal{A} \times \Theta \to \mathcal{U}$ a map representing a parametric class of operators with parameters $\theta \in \Theta$. Given a discrete refinement $(\Omega_J)_{J=1}^{\infty}$ of the domain $\Omega \subset \mathbb{R}^d$, we say $\mathcal{G}$ is discretization-invariant if there exists a sequence of maps $\hat{\mathcal{G}}_1, \hat{\mathcal{G}}_2, \ldots$, where $\hat{\mathcal{G}}_J : \mathbb{R}^{Jd} \times \Theta \to \mathcal{U}$, such that

$$\lim_{J \to \infty} \sup_{a \in K} \left\| \mathcal{G}\left(a|_{\Omega_J} ; \theta\right) - \mathcal{G}(a; \theta) \right\|_{\mathcal{U}} = 0,$$

for any $\theta \in \Theta$ and any compact set $K \subset \mathcal{A}$.

Generally speaking, discretization invariance defined in Kovachki et al. (2021) refers to the property that a neural operator taking discrete data as input converges to the continuum operator taking continuous functions as input as the discretization becomes finer. However, this does not necessarily imply that the neural operator's performance is consistent regardless of the input data's discretization, which is a common misconception within the operator learning community. ***We note that this property implies convergence to the continuum, rather than the ability to achieve cross-resolution tasks.*** Instead, throughout this paper, we will refer to discretization invariance as the property that the neural operator's parameters remain decoupled from the resolution of the data **and** that the neural operator can take inputs of different discretizations directly without sacrificing its performance. Many in the operator learning community interpret this as the meaning of discretization invariance, often conducting zero-shot super-resolution tests in their research. To avoid confusion with the discretization (or resolution) invariance as defined in Kovachki et al. (2021), we will use the term "cross-resolution". To this end, we introduce DMEs, a key concept for cross-resolution pipelines.

**Definition 4.3.** Let $\Theta \subseteq \mathbb{R}^p$ be a finite dimensional parameter space and $\mathcal{G} : \mathcal{A} \times \Theta \to \mathcal{U}$ a map representing a parametric class of Fourier neural operators with parameters $\theta \in \Theta$. Given a uniform

discrete refinement $(\Omega_J)_{J=1}^\infty$ of the domain $\Omega \subset \mathbb{R}^d$, the *discretization mismatch error* between the Fourier neural operator taking two different discretizations of the function $a$ is defined as

$$E_{MN} := \left\| \mathcal{G}_\theta \left( a|_{\Omega_M} \right) - \mathcal{G}_\theta \left( a|_{\Omega_N} \right) \right\|_{\mathcal{U}}.$$

## 4.2 THE ACCUMULATION OF DISCRETIZATION MISMATCH ERROR

To provide a detailed analysis of discretization errors in FNOs, we formally and mathematically outline the full FNO architecture in Appendix A.1 and present Proposition 4.4.

**Proposition 4.4.** *Let $\Theta \subseteq \mathbb{R}^p$ be a finite dimensional parameter space, $\mathcal{A} = H^s \left( \mathbb{T}^d; \mathbb{R}^{d_a} \right)$, $\mathcal{U} = H^{s'} \left( \mathbb{T}^d; \mathbb{R}^{d_u} \right)$, and $\mathcal{G} : \mathcal{A} \times \Theta \to \mathcal{U}$ a map representing a parametric class of Fourier neural operators with parameters $\theta \in \Theta$ with $L$ Fourier layers. Consider $\sigma(x)$ as the activation function that is (globally) $\omega$-Lipschitz continuous for all layers. Given a discretization $a|_{\Omega_N}$, then the discretization mismatch error between the Fourier neural operator taking $a|_{\Omega_N}$ and another discretization $a|_{\Omega_M}$, given by $E_{MN} = \left\| \mathcal{G}_\theta \left( a|_{\Omega_M} \right) - \mathcal{G}_\theta \left( a|_{\Omega_N} \right) \right\|$ with $M > N$, will increase as $M$ increases. Additionally, the discretization mismatch error might increase as $L$ and $\omega$ increase.*

Proposition 4.4 indicates that DMEs propagate and accumulate across the intermediate Fourier layers, and as a straightforward extension from the continuity property of $\mathcal{G}$, the DMEs also propagate over time in autoregressive tasks. The training resolution can be $M$ or $N$, so this bound suggests that as long as the inference resolution differs from that of the training, it will suffer from DMEs; nevertheless, it is a super-resolution or sub-resolution (lower resolution) task. We demonstrate this empirically in Sec. 5.1. The proof for this proposition and formal statements on the upper bound is given in Appendix B. We experimentally verify our findings in Sec. 5.1. These findings suggest that neural operators may struggle to perform cross-resolution tasks. Therefore, we propose the Cross-Resolution Operator-learning Pipeline to address this issue.

## 4.3 CROP: MINIMIZING THE DISCRETIZATION MISMATCH ERROR

As shown in Sec. 4.2, the DMEs arise in neural operators, limiting their ability to perform cross-resolution tasks. To address this, we propose the CROP lifting and projection layers:

$$P_{\text{CROP}} : H^s \left( \mathbb{T}^d; \mathbb{R}^{d_a} \right) \to \mathcal{B}_w \left( \mathbb{T}^d; \mathbb{R}^{d_0} \right) \quad \text{and} \quad Q_{\text{CROP}} : \mathcal{B}_w \left( \mathbb{T}^d; \mathbb{R}^{d_L} \right) \to H^s \left( \mathbb{T}^d; \mathbb{R}^{d_a} \right).$$

The CROP lifting layer, $P_{\text{CROP}}$, maps the input function to a band-limited, usually high-dimensional, latent feature function through a global convolution with a band-limited convolution kernel $\mathcal{C}(x-y)$. Hence, $v_0(x) = \int_{\mathbb{T}^d} \mathcal{C}(x-y)a(x)dy$, and the resulting latent feature function $v_0(x)$ will be band-limited as well. As we consider a band-limited function space, we can fix the discretization of the latent feature function in accordance with the band-limit we choose. Since the discretization is fixed for the latent feature functions, any intermediate neural operator architecture $\mathcal{L}$ can be chosen to learn the mapping between latent feature functions, including those that are mesh-dependent:

$$\mathcal{L} : \mathcal{B}_w \left( \mathbb{T}^d; \mathbb{R}^{d_0} \right) \mapsto \mathcal{B}_w \left( \mathbb{T}^d; \mathbb{R}^{d_L} \right).$$

Moreover, because the latent discretizations are fixed, CROP does not suffer from the DME, and we can interpret the intermediate neural operator as an operation on these discretizations. As a result, it will not introduce high-frequency components beyond the established band-limit that the discretization can accommodate. Therefore, the intermediate neural operator satisfies the Continuous-Discrete Equivalence (Raonic et al., 2023).

Similarly, the CROP projection layer, $Q_{\text{CROP}}$, consists of a global band-limited kernel convolution combined with a fully connected point-wise network to map the band-limited latent feature function to an output function with frequencies that can exceed the band-limit in the appropriate function space. The high-frequency information beyond the band-limit is learned in the CROP projection layer through a fully connected point-wise network, potentially augmented by a residual connection to the input, as tolerated by the refined discretization.

*Remark* 4.5. Let $v(x) \in \mathcal{B}_w \left( \mathbb{T}^d; \mathbb{R}^{d_L} \right)$ be a band-limited function, and let an analytic non-linear function $\sigma(\cdot)$ be the fixed activation function in the fully connected point-wise network. Then the resulting function may not be band-limited anymore. Additionally, the high frequencies of the resulting function is determined by both the fully connected point-wise network and the learned latent feature function $v(x)$.

The proof of this remark is given in Appendix C. This remark suggests that the high-frequency components beyond the band-limit are determined by both the fully connected point-wise network and $v$.

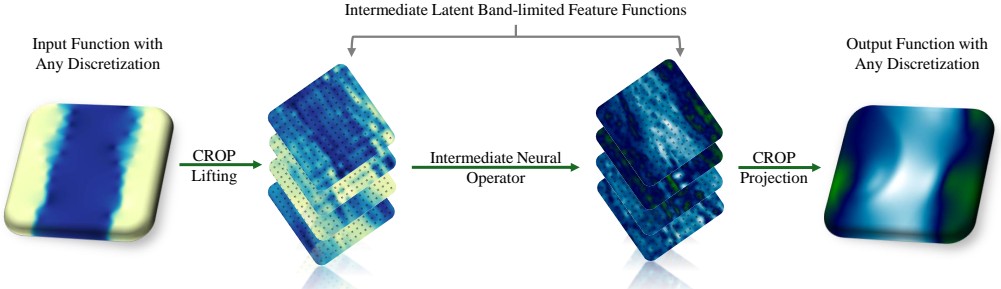

Figure 2: Cross-Resolution Operator learning Pipeline: Firstly, a CROP lifting layer maps the input parametric function to a latent feature function in a band-limited function space with a fixed discretization. Consequently, an intermediate neural operator that operates on band-limited function spaces is applied to process the latent feature function. **This intermediate neural operator does not necessarily have to be discretization invariant, so we can choose models of local features such as the U-Net.** Finally, a CROP projection layer will map the latent feature function to final output function in the appropriate function space. A Fully connected, point-wise network is incorporated in the CROP projection layers to learn higher frequency information beyond the band-limits.

The intermediate neural operator is tasked to learn an appropriate latent feature function $v$ to capture high-frequency components in conjunction with the fully connected point-wise network. The key ideas of CROP is to have the latent feature functions to be in a band-limited function space so that they can be processed by an intermediate neural operator that can be more freely chosen; for example, we can use mesh-dependent architectures such as CNNs. This process, however, will "CROP" out high frequencies, we utilize the CROP projection layer to recover such frequencies. We provide a spectral comparison between FNO and CROP in Appendix C.1. The CROP lifting and projection layers are able to capture global information efficiently; thus, we can select an intermediate neural operator of the local nature, such as U-Nets, without sacrificing the ability to learn global features. Overall, this framework is more robust for cross-resolution tasks by having an intermediate neural operator that maps between band-limited function spaces and we can choose appropriate intermediate neural operators to efficiently learn the multi-spatio-scale nature of many problems in physical modeling (Gupta & Brandstetter, 2022).

## 5 EXPERIMENTAL RESULTS

In this section, we conduct experiments to substantiate our claims and demonstrate the effectiveness of our proposed CROP components. In Sec. 5.1, we conduct experiments on the incompressible Navier-Stokes equation to showcase the accumulation of DMEs and verify our CROP components' ability to perform cross-resolution tasks. In Sec. 5.2, we conduct experiments to demonstrate the efficient multi-spatio-scale learning capability of our CROP components.

### 5.1 CROSS-RESOLUTION TESTS

In this example, we demonstrate the DME in neural operators as discussed in Sec. 4.2 and then explore the ability of CROP components for cross-resolution tasks.

**Description.** We consider the 2D Navier-Stokes equation for a viscous, incompressible fluid in vorticity form, as described in Li et al. (2021):

$$\partial_t w(x,t) + u(x,t) \cdot \nabla w(x,t) = \nu \Delta w(x,t) + f(x), \quad x \in (0,1)^2, t \in (0,T]$$
$$\nabla \cdot u(x,t) = 0, \quad x \in (0,1)^2, t \in [0,T] \qquad (3)$$
$$w(x,0) = w_0(x), \quad x \in (0,1)^2$$

where $u \in C\left([0,T]; H_{\mathrm{per}}^r\left((0,1)^2; \mathbb{R}^2\right)\right)$ for any $r > 0$ is the velocity field, $w = \nabla \times u$ is the vorticity, $w_0 \in L_{\mathrm{per}}^2\left((0,1)^2; \mathbb{R}\right)$ is the initial vorticity, $\nu = 1\mathrm{e} - 3$ is the viscosity coefficient, and $f \in L_{\mathrm{per}}^2\left((0,1)^2; \mathbb{R}\right)$ is the forcing function. We are interested in learning the operator that maps the vorticity up to time 10 to the vorticity up to a later time $T = 50$ in an auto-regressive manner for all the intermediate time steps with a step size of 1 following Li et al. (2021). More details on the experimental setups can be found in Appendix D.1.1 and D.1.2.

### 5.1.1 THE ACCUMULATION OF DISCRETIZATION MISMATCH ERROR

First, we recognize that neural operators may not be robust across different resolutions. In Figure 3 (a), we present results from a well-trained FNO on 200 testing input parametric functions and plot the average relative $\ell_2$ error under different resolutions; we can clearly observe the impacts of the DME. Even at the very first time step when the error has not propagated through time yet, inferences at $128 \times 128$ (0.66%) are three times as high as those at the training resolution (0.22%), and inferences at other resolutions are even worse.

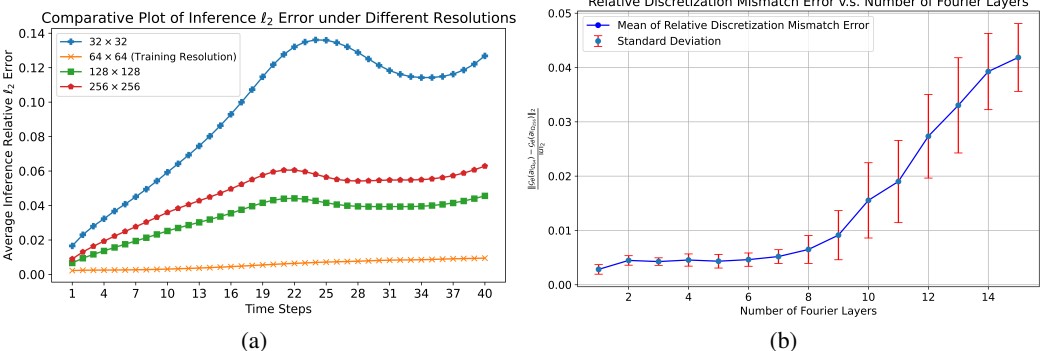

(a)        (b)

Figure 3: (a): The relative $\ell_2$ error during inference for the Navier-Stokes equation across different input resolutions. We can clearly observe the effects of DME on performance, even at the very first time step. Unsurprisingly, this effect propagates through the auto-regressive time steps. (b): The growth in relative DME with respect to increasing numbers of layers. This highlights our observation that DMEs propagate through the Fourier layers.

As shown in Figure 3 (b), there is a clear trend: as the network stacks more layers, the DME increases. As significant discretization errors exist within the network architecture, their impact on the overall performance of super-resolution tasks is profound. Therefore, based on both experimental observations and mathematical analysis, we conclude that FNO is not suited for cross-resolution tasks (inference on resolutions different from that of the training). However, we emphasize again that the formulation of FNO is revolutionary and lays the foundation for function space mappings. Philosophically, since the training data lacks high-resolution information, it remains unclear whether neural operators can learn what they have not seen (high-resolution information) to achieve zero-shot super-resolution without the inclusion of specific designs, such as transfer learning, physics-informed loss (Li et al., 2022; Wang et al., 2021), conservation laws (Liu et al., 2024), symmetries (Helwig et al., 2023; Gao et al., 2024), etc.. However, we believe that FNO offers great potential to achieve data-driven super-resolution tasks; we will discuss this further in Appendix E.

### 5.1.2 CROP: MINIMIZING THE DISCRETIZATION MISMATCH ERROR

We demonstrate that CROP can achieve robust performance across different resolutions of inference. We compare the CROP-enhanced FNO model with the plain FNO and the FNO with anti-alias activation functions (Raonic et al., 2023).

Table 2: The mean and standard deviation of relative $\ell_2$ error over 10 runs during inference are reported for the Navier-Stokes Equation with varying resolutions of input parametric functions. CROP is consistent across all resolutions. Baseline models continue to show a strong bias toward the training resolution. The DME is obviously shown when inferred on data with different resolutions.

| | $256 \times 256$ | $128 \times 128$ | $64 \times 64^{\diamondsuit}$ | $32 \times 32$ |
|---|---|---|---|---|
| FNO | $4.65(\pm 1.47)$ | $3.35(\pm 0.97)$ | $0.58(\pm 0.04)$ | $9.45(\pm 2.92)$ |
| Alias-free FNO ($2\times$) | $6.91(\pm 2.73)$ | $6.18(\pm 3.01)$ | $0.61(\pm 0.04)$ | $39.74(\pm 14.11)$ |
| Alias-free FNO (Fixed) | $6.95(\pm 2.60)$ | $7.40(\pm 3.49)$ | $0.61(\pm 0.04)$ | $42.08(\pm 15.03)$ |
| CNO (with interpolation) | $4.17(\pm 1.46)$ | $2.95(\pm 1.06)$ | $1.98(\pm 0.14)$ | $7.69(\pm 1.69)$ |
| CROP (ours) | $\mathbf{0.54(\pm 0.05)}$ | $\mathbf{0.54(\pm 0.05)}$ | $\mathbf{0.54(\pm 0.05)}$ | $\mathbf{0.54(\pm 0.05)}$ |

$\diamondsuit$ Training resolution is $64 \times 64$.

We present the results in Table 5.1.2 as percentages. It is clear that FNOs exhibit a strong bias toward the training resolution, regardless of whether upsampling and downsampling are used to

mitigate aliasing issues (Raonic et al., 2023). We additionally include the cross-resolution results for CNO (Raonic et al., 2023) with downsampling and bicubic interpolation to illustrate that relying on explicit up/downsampling may degrade performance. Therefore, we assert that alias-free activation functions cannot fully enable FNO to succeed in cross-resolution tasks. The major challenge we need to solve is the discretization error inherent in the realization of neural operators at a discrete level. Clearly, CROP minimizes the DME and achieves the best results on cross-resolution tasks. Moreover, CROP exhibits superior performance compared to other baselines even when inferring at the same resolution of the training data. This can be attributed to the fact that CROP is continuous-discrete-equivalent; FNO truncates some high-frequency modes in each layer for efficiency and reduction of the number of parameters, but this truncation will break continuous-discrete-equivalence. Moreover, such truncations are necessary if we aim to train at a lower resolution and infer at a different resolution. CROP strictly adheres to the band-limits in accordance with the truncation; thus, CROP improves the overall results. Additionally, as CROP does not suffer from DMEs in any autoregressive time step, CROP is stable for long-term roll-outs in terms of cross-resolution abilities.

## 5.2 LEARNING CAPABILITY TEST

In this example, we explore the learning capabilities of CROP components. We demonstrate that CROP achieves superior performance compared to the baselines. This performance gain can be attributed to factors such as exceptional multi-spatio-scale abilities while being free from aliasing and DMEs, as outlined in Sec. 4.3. Additionally, we include the results of cross-resolution tests in Appendix D.5 to showcase CROP's cross-resolution capabilities.

### 5.2.1 NAVIER-STOKES EQUATION WITH HIGH REYNOLDS NUMBER

**Description.** We examine the same Navier-Stokes equation discussed in Section 5.1, but with an increased Reynolds number. High Reynolds numbers pose significant learning challenges due to the turbulent characteristics and small-scale features of the flow (Liu-Schiaffini et al., 2024). We conduct experiments at Reynolds numbers of $5,000$ and $10,000$, respectively. As higher Reynolds numbers lead to more complex problems, we utilize a larger training dataset for $Re = 10,000$. More details can be found in Appendix D.1.3.

Table 3: Results on the Navier-Stokes equation with high Reynolds numbers are presented. The mean and standard deviation of the relative $\ell_2$ error over 10 inference runs are provided. Timing results are also included as a reference. Clearly, CROP outperforms all baselines.

|  |  |  |  | $Re = 5000$ | $Re = 10000$ |
|---|---|---|---|---|---|
|  | #Par. (M) | BPS | IT | Test (%) | Test (%) |
| FNO | $2.27^*$ | 18.47 | 3.39 | 7.25($\pm$0.38) | 8.35($\pm$0.55) |
| CNO | 2.61 | 1.64 | 9.73 | 5.77($\pm$0.19) | 4.66($\pm$0.07) |
| U-Net | 7.76 | 34.54 | 1.56 | 8.96($\pm$0.06) | 8.75($\pm$0.22) |
| ResNet | 23.50 | 2.01 | 6.73 | 57.79($\pm$2.11) | 53.01($\pm$0.96) |
| DeepONet | 4.89 | 50.35 | 0.89 | 16.93($\pm$0.40) | 17.60($\pm$0.37) |
| CROP (ours) | $4.03^*$ | 12.70 | 4.69 | **3.84($\pm$0.09)** | **4.06($\pm$0.22)** |

BPS (Batch per Second): The number of batches processed per second during training.

IT (Inference Time): The time, in milliseconds, it takes to infer on a batch of 16 samples.

$^*$: Each complex parameter (which is twice the size of a real parameter) is counted as 2.

**Results and Discussion.** The results clearly demonstrate that CROP exhibits exceptional learning capabilities, effectively capturing both fine-scale and global features. While Fourier Neural Operator (FNO) performs well at low Reynolds numbers, as shown in Sec. 5.1, it tends to over-smooth fine-scale details, leading to degraded performance. Although the Convolutional Neural Operator (CNO) can learn global features through its U-path, it excels primarily in local contexts due to its convolutional nature, making it less effective at capturing long-range dependencies compared to our CROP components. Similarly, U-Net and ResNet also struggle with learning long-range dependencies. We recognize that DeepONet is a groundbreaking framework that performs well in various applications; however, like FNO, it may also over-smooth fine-scale details. Overall, CROP's ability to effectively learn complex patterns results in superior performance compared to the baselines. An interesting phenomenon is that FNO performs worse while CNO performs better under higher Reynolds numbers. Our hypothesis is that as the Reynolds number increases, fine-scale details become more crucial to learn due to the increased chaos. CROP consistently performs well under both Reynolds numbers due to the multi-spatio-scale learning capabilities it offers.

### 5.2.2 Darcy Flow, Helmholtz, and Poisson Equations

**Description: Darcy Flow.** We consider the steady-state of the 2D Darcy flow equation from Li et al. (2021); Hasani & Ward (2024). We learn both the nonlinear operator mapping the diffusion coefficient to the solution function, $a \mapsto u$, following the same setup and data as in Li et al. (2021), and the operator mapping the forcing function to the solution function, $f \mapsto u$, following the same generation setup in Hasani & Ward (2024).

**Description: Helmholtz.** We consider the 2D Helmholtz equation from De Hoop et al. (2022). We learn the nonlinear operator mapping the wave-speed field to the solution function, $c \mapsto u$, following the same setup and data as in De Hoop et al. (2022) with normalization on the data.

**Description: Poisson.** We consider the 2D Poisson equation from Raonic et al. (2023) and learn the operator mapping the forcing function to the solution function, $f \mapsto u$, following the same setup and data as in Raonic et al. (2023). More details about these equations, data generation, and setups are provided in Appendix D.

Table 4: Results for the Darcy flow, Helmholtz, and Poisson equations are presented. Nonlinear Darcy denotes the non-linear mapping $a \mapsto u$. It is evident that CROP achieves competitive performance with multi-spatio-learning capabilities while maintaining the cross-resolution property.

|  | Nonlinear Darcy | Darcy Flow | Helmholtz | Poisson |
|---|---|---|---|---|
| FNO | 0.68($\pm$0.03) | 1.58($\pm$0.14) | 0.68($\pm$0.01) | 7.21($\pm$0.74) |
| CNO | 0.59($\pm$0.04) | 1.32 $\pm$ (0.33) | 0.87($\pm$0.02) | 1.67($\pm$0.56) |
| U-Net | 1.34($\pm$0.03) | 2.68($\pm$0.17) | 1.51($\pm$0.06) | 1.40($\pm$0.21) |
| ResNet | 7.26($\pm$0.59) | 9.85($\pm$0.68) | 14.86($\pm$0.53) | 5.64($\pm$0.34) |
| DeepONet | 2.99($\pm$0.10) | 1.36($\pm$0.16) | 3.03($\pm$0.08) | 12.26($\pm$1.48) |
| CROP (ours) | **0.51($\pm$0.01)** | **1.11($\pm$0.08)** | **0.52($\pm$0.02)** | **0.80($\pm$0.13)** |

**Results and Discussion.** The findings clearly indicate that CROP demonstrates outstanding learning abilities because we can freely choose an intermediate operator while maintaining properties such as resolution-invariance, discrete-continuous equivalence, and freedom from DMEs. This flexibility enables us to select a suitable intermediate neural operator that focuses on local details. For the Darcy tasks, as this is a relatively easy learning task, all models, except ResNet, perform well; however, CROP still shows improvement over all baselines. We additionally include the cross-resolution performance for CROP and FNO on nonlinear Darcy in Appendix D.2. A similar trend is observed for the Helmholtz equation, where all models perform well except ResNet, and CROP outperforms all baselines. For the Poisson equation, we note that architectures with local characteristics, including CNO, U-Net, and ResNet, all perform better than FNO. CROP, as a multi-spatio-scale learning paradigm with both local and global learning capabilities, outperforms all baselines by attending to global patterns while focusing on local structures.

## 6 Conclusion

In this work, we have explored the challenges posed by discretization mismatch errors (DMEs) in neural operators. Despite FNO's advantages in formulating the kernels in the Fourier domain to capture global features and learn dynamic physical phenomena, its performance is not consistent across different resolutions. Moreover, FNO tends to over-smooth fine-scale details. We identified the sources of these discrepancies and introduced a Cross-Resolution Operator-learning Pipeline (CROP) designed to mitigate DMEs. Our proposed method not only succeeds in cross-resolution tasks but also outperforms baselines in learning complex dynamics, as it upholds continuous-discrete equivalence and efficiently learns both global and local features through CROP designs.

**Limitations and Future Work.** First, like FNO, CNO, and other CNN-based architectures, our CROP components perform optimally with rectangular domains and uniform grids. We acknowledge that this limitation may affect the applicability of CROP. Efforts have been made to address these drawbacks in FNO and CNN-based architectures (Liu et al., 2023; Gao et al., 2021). While these concepts may be adapted for CROP, we plan to explore this in the future. Second, although our method can capture high-frequency information using both the learned feature function and the fully-connected pointwise network, further investigation into the expressive power and universality of these components would enhance our understanding and inform better designs for intermediate neural operators and activation functions in operator learning.

## 7 ACKNOWLEDGMENT

This work has used the computational equipment supported by the US Army Research Office under the award W911NF-20-10159. Y. Liu and W. Gao would like to thank Dr. Xiaolin Li at Stony Brook University for providing this computational equipment.

## ETHICS STATEMENT

This research focuses on the development and evaluation of neural operators, which are mathematical and deep learning tools designed to model complex systems. The study does not involve human subjects, personal data, or sensitive information that could raise privacy, security, or fairness concerns. No potential conflicts of interest, legal compliance issues, or harmful applications have been identified in this work.

## REPRODUCIBILITY

All baseline models used in this work were adopted with minimal or no modifications from their original implementations. The datasets utilized are either open-source or generated using publicly available code, as described in detail in the appendix. We provide the source code, datasets, pre-trained models, and configuration necessary to replicate the key experiments at `https://github.com/wenhangao21/ICLR25-CROP`. Comprehensive instructions for training and evaluation will also be included. Additionally, all theoretical claims are accompanied by proofs in the appendix, which have been empirically verified through experiments.

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

## A    ACCUMULATION OF DISCRETIZATION ERROR IN FNO

### A.1    MATHEMATICAL FORMULATION OF THE FNO ARCHITECHTURE

To give a detailed analysis of discretization errors in FNOs, we first formally and mathematically lay out the most common FNO architecture. Let $\mathcal{A}\left(\Omega, \mathbb{R}^{d_a}\right)$ and $\mathcal{U}\left(\Omega, \mathbb{R}^{d_u}\right)$ be two appropriate function spaces defined on bounded domain $\Omega \subset \mathbb{R}^d$. The Fourier neural operator $\mathcal{G} : \mathcal{A} \to \mathcal{U}$ is defined as a compositional mapping between function spaces:

$$\mathcal{G}(a) := Q \circ \mathcal{L}_L \circ \mathcal{L}_{L-1} \circ \cdots \circ \mathcal{L}_1 \circ P(a). \tag{4}$$

The input function $a \in \mathcal{A}$ is lifted to the latent space of $R^{d_{v_0}}$-valued (usually $d_{v_0} > d_a$) functions through a lifting layer acting locally:

$$P : \left\{ a : \Omega \to \mathbb{R}^{d_a} \right\} \mapsto \left\{ v_0 : \Omega \to \mathbb{R}^{d_{v_0}} \right\}.$$

Lifting layer usually is implemented as a linear layer represented by a matrix $P \in \mathbb{R}^{d_v \times d_a}$ or as a point-wise multi-layer perceptron (1 by 1 convolution layers) with activation function $\sigma$.

Then the result goes through $L$ Fourier layers:

$$\mathcal{L}_\ell(v)(x) = \sigma \left( W_\ell v(x) + b_\ell(x) + \mathcal{K}_\ell v(x) \right), \quad \ell = 1, 2, \ldots, L, \tag{5}$$

where, $\sigma$ is a non-linear activation function, $W_\ell$ acts locally and can be represented by a matrix $\in \mathbb{R}^{d_{v_{\ell-1}} \times d_{v_\ell}}$, $b_\ell(x) \in \mathcal{U}\left(\Omega; \mathbb{R}^{d_v}\right)$ is the bias function (usually a constant function for easy implementation), and

$$\mathcal{K}_\ell v(x) = \mathcal{F}^{-1}\left( P_\ell(k) \cdot \mathcal{F}(v)(k) \right)(x)$$

is a linear but non-local convolution operator carried out in the Fourier space with $P_\ell : \mathbb{Z}^d \to \mathbb{C}^{d_v \times d_v}$ being the Fourier coefficients of the convolution kernels. We will denote the output of the $\ell$-th Fourier layer by $v_\ell$.

The output function is obtained from the projection layer acting locally, similar to the lifting layer:

$$Q : \left\{ v_L : \Omega \to \mathbb{R}^{d_{v_L}} \right\} \mapsto \left\{ u : \Omega \to \mathbb{R}^{d_u} \right\}.$$

For simplicity, we will assume that all the inputs and outputs to the Fourier layers have the same channel dimension, i.e., $d_{v_0} = d_{v_1} = \ldots d_{v_L} = d_c$.

In practice, the FNO, as well as all the intermediate layers, takes discretizations of functions as inputs and produces discretizations of functions as outputs. Therefore, this discrete implementation is termed the Pseudo Fourier Neural Operators; readers are referred to Kovachki et al. (2021) for more details.

## B    PROOF TO PROPOSITION 4.4

We first present the following lemma for the discretization error of $L$ Fourier layers and with only a single hidden channel. We will later extend this result to the full FNO architecture and with $d_c$ hidden channels in general. In this section, all norms are $\ell_2$ norms unless specified otherwise.

**Lemma B.1.** *Let $\Theta \subseteq \mathbb{R}^p$ be a finite-dimensional parameter space, and let $s > 2$ be an integer. Consider $\mathcal{L} : H^s\left(\mathbb{T}; \mathbb{R}\right) \times \Theta \to H^s\left(\mathbb{T}; \mathbb{R}\right)$ being the composition of $L$ parametric Fourier layers with parameters $\theta \in \Theta$ and a hidden channel dimension of one. Assume the activation function $\sigma : \mathbb{R} \to \mathbb{R}$ is $\omega$-Lipschitz continuous. Given a discretization $v|_{\Omega_N}$, the discretization mismatch error between the Fourier layers $\mathcal{L}$ taking $v|_{\Omega_N}$ and another discretization $v|_{\Omega_M}$ is bounded by*

$$E_{NM} = \|\mathcal{L}\left(v|_{\Omega_N}\right) - \mathcal{L}\left(v|_{\Omega_M}\right)\|_{L^2(\mathbb{T})}$$

$$\leq \omega^L \left( \prod_{\ell=1}^{L} C_\ell \right) \sum_{\xi=0}^{N-1} \left[ \frac{2\pi|N-M|}{NM} \xi \left| \sum_{n=1}^{N} v(x_n) n e^{-i2\pi \frac{n}{N}\xi} \right| + \left| \sum_{n=N+1}^{M} v(x_n) e^{-i2\pi \frac{n}{M}\xi} \right| \right], \tag{6}$$

*where without the loss of generality we assume $M > N$, $C_\ell = \|\mathcal{W}_\ell\| + C_{\kappa_\ell}$, with $\|\mathcal{W}_\ell\|$ being the operator norm of the linear operator $\mathcal{W}_\ell$ in the $\ell$-th layer, and $C_{\kappa_\ell}$ is a constant depending on the $\ell$-th layer's kernel $\kappa_\ell$.*

*Proof.* We aim to establish a bound on the discretization mismatch error $E_{MN}$ between the outputs of $\mathcal{L} = \mathcal{L}_L \circ \mathcal{L}_{L-1} \circ \cdots \circ \mathcal{L}_1$ applied to the input function at different discretizations $\Omega_N$ and $\Omega_M$, where each layer $\mathcal{L}_\ell$ is defined as $\mathcal{L}_\ell(f(x)) = \sigma\left(\mathcal{W}_\ell f(x) + b_\ell(x) + \mathcal{K}_\ell(f(x))\right)$. We view $v_N$ and $v_M$ as functions through their Fourier coefficients that are approximated numerically.

Applying $\mathcal{L}_1$ to $v_N$ and $v_M$, we have:

$$\mathcal{L}_1(v_N)(x) = \sigma\left(\mathcal{W}_1 v_N(x) + b_1(x) + \mathcal{K}_1(v_N)(x)\right),$$

$$\mathcal{L}_1(v_M)(x) = \sigma\left(\mathcal{W}_1 v_M(x) + b_1(x) + \mathcal{K}_1(v_M)(x)\right).$$

Let the difference between the outputs be $\Delta_1(x) = \mathcal{L}_1(v_N)(x) - \mathcal{L}_1(v_M)(x)$.

By the Lipschitz continuity of $\sigma$, we have:

$$\begin{aligned}
|\Delta_1(x)| &\leq \omega \left|(\mathcal{W}_1 v_N(x) + \mathcal{K}_1(v_N)(x)) - (\mathcal{W}_1 v_M(x) + \mathcal{K}_1(v_M)(x))\right| \\
&= \omega \left(|\mathcal{W}_1(v_N(x) - v_M(x))| + |\mathcal{K}_1(v_N)(x) - \mathcal{K}_1(v_M)(x)|\right).
\end{aligned} \tag{7}$$

Assuming that $\mathcal{W}_1$ is a bounded linear operator with operator norm $\|\mathcal{W}_1\|$, we have:

$$|\mathcal{W}_1(v_N(x) - v_M(x))| \leq \|\mathcal{W}_1\| |v_N(x) - v_M(x)|.$$

Moreover, the convolution difference is:

$$\mathcal{K}_1(v_N)(x) - \mathcal{K}_1(v_M)(x) = \int_{\mathbb{T}} \kappa_1(x - y)\left(v_N(y) - v_M(y)\right) dy.$$

The difference after the $\ell$-th layer is:

$$\Delta^{(\ell)} = v_N^{(\ell)} - v_M^{(\ell)}.$$

Using the Lipschitz continuity of $\sigma$ with constant $\omega$ and the boundedness of $\mathcal{W}_\ell$ and $\mathcal{K}_\ell$, we have:

$$\left\|\Delta^{(\ell)}\right\|_{L^2(\mathbb{T})} \leq \omega\left(\|\mathcal{W}_\ell\| + C_{\kappa_\ell}\right)\left\|\Delta^{(\ell-1)}\right\|_{L^2(\mathbb{T})},$$

where $C_{\kappa_\ell} = \int_{\mathbb{T}} |\kappa_\ell(z)| dz$. By induction over $L$ layers, it follows that:

$$\left\|\Delta^{(L)}\right\|_{L^2(\mathbb{T})} \leq \omega^L \left(\prod_{\ell=1}^{L}\left(\|\mathcal{W}_\ell\| + C_{\kappa_\ell}\right)\right)\left\|\Delta^{(0)}\right\|_{L^2(\mathbb{T})}.$$

We now estimate $\left\|\Delta^{(0)}\right\|_{L^2(\mathbb{T})}$. Let the difference between the Fourier coefficients be

$$\widehat{\Delta}(\xi) = \widehat{v}^N(\xi) - \widehat{v}^M(\xi) = \sum_{n=1}^{N} v(x_n)\left(e^{-i2\pi\frac{n}{N}\xi} - e^{-i2\pi\frac{n}{M}\xi}\right) - \sum_{n=N+1}^{M} v(x_n)e^{-i2\pi\frac{n}{M}\xi},$$

where $e^{-i2\pi\frac{n}{N}\xi} - e^{-i2\pi\frac{n}{M}\xi}$ can be approximated using a Taylor expansion:

$$e^{-i2\pi\frac{n}{N}\xi} - e^{-i2\pi\frac{n}{M}\xi} \approx -i2\pi n\left(\frac{1}{N} - \frac{1}{M}\right)\xi e^{-i2\pi\frac{n}{N}\xi} = -i2\pi n\frac{N-M}{NM}\xi e^{-i2\pi\frac{n}{N}\xi}.$$

It follows that

$$\left|\widehat{\Delta}(\xi)\right| \leq 2\pi\frac{|N-M|}{NM}\xi\left|\sum_{n=1}^{N} v(x_n)n e^{-i2\pi\frac{n}{N}\xi}\right| + \left|\sum_{n=N+1}^{M} v(x_n)e^{-i2\pi\frac{n}{M}\xi}\right|.$$

Using Parseval's identity, we can sum over the relevant frequencies:

$$\left\|\Delta^{(0)}\right\|_{L^2(\mathbb{T})} \leq \sum_{\xi=0}^{N-1}\left(\left|\widehat{\Delta}(\xi)\right| + \left|\widehat{\Delta}(-\xi)\right|\right).$$

Substituting the estimate of $\left|\widehat{\Delta}(\xi)\right|$, we obtain:

$$\left\|\Delta^{(0)}\right\|_{L^2(\mathbb{T})} \leq \sum_{\xi=0}^{N-1} \left[ 2\pi \frac{|N-M|}{NM} \xi \left| \sum_{n=1}^{N} v(x_n) n e^{-i2\pi \frac{n}{N}\xi} \right| + \left| \sum_{n=N+1}^{M} v(x_n) e^{-i2\pi \frac{n}{M}\xi} \right| \right].$$

Therefore, combining the above estimates and simplify, we have:

$$E_{NM} = \left\| \mathcal{L}\left(v|_{\Omega_N}\right) - \mathcal{L}\left(v|_{\Omega_M}\right) \right\|_{L^2(\mathbb{T})}$$
$$\leq \omega^L \left( \prod_{\ell=1}^{L} C_\ell \right) \sum_{\xi=0}^{N-1} \left[ \frac{2\pi |N-M|}{NM} \xi \left| \sum_{n=1}^{N} v(x_n) n e^{-i2\pi \frac{n}{N}\xi} \right| + \left| \sum_{n=N+1}^{M} v(x_n) e^{-i2\pi \frac{n}{M}\xi} \right| \right],$$

(8)

where without the loss of generality we assume $M > N$, $C_\ell = \|\mathcal{W}_\ell\| + C_{\kappa_\ell}$, with $\|\mathcal{W}_\ell\|$ being the operator norm of the linear operator $\mathcal{W}_\ell$ in the $\ell$-th layer. $\qquad \square$

**Intepretation of this bound:** When $N = M$, clearly this bound will be 0. Without loss of generality, we assume $M > N$, when $N$ is fixed, $\frac{|N-M|}{NM}$ gradually increases to 1, and as $M$ increases, this bound increases. Moreover, $\sum_{\xi=0}^{N-1} \left| \sum_{n=N+1}^{M} v(x_n) e^{-i2\pi \frac{n}{M}\xi} \right|$ behaves as a term that quantifies the difference between the input functions. The errors will propagate through the layers.

Now, we are ready to prove Proposition 4.4, which has been written to help readers better understand our results. We begin by proving the following lemma concerning the error bound of the discretization error for the full FNO, which implies Proposition 4.4.

**Lemma B.2.** *Let* $\Theta \subseteq \mathbb{R}^p$ *be a finite-dimensional parameter space,* $\mathcal{A} = H^s\left(\mathbb{T}^d; \mathbb{R}^{d_a}\right)$, $\mathcal{U} = H^{s'}\left(\mathbb{T}^d; \mathbb{R}^{d_u}\right)$, *and* $\mathcal{G} : \mathcal{A} \times \Theta \to \mathcal{U}$ *a map representing a parametric class of Fourier neural operators with parameters* $\theta \in \Theta$ *with* $L$ *Fourier layers and* $d_c$ *hidden channels. Consider* $\sigma(x)$ *as the activation function that is (globally)* $\omega$*-Lipschitz continuous. Given a discretization* $a|_{\Omega_N}$, *then the discretization mismatch error between the Fourier neural operator taking* $a|_{\Omega_N}$ *and another discretization* $a|_{\Omega_M}$, *given by*

$$E_{NM} = \left\| \mathcal{G}\left(a|_{\Omega_N}\right) - \mathcal{G}\left(a|_{\Omega_M}\right) \right\|_{L^2(\mathbb{T}^d)}$$
$$\leq C\omega^L \left( \prod_{\ell=1}^{L} C_\ell \right) \sum_{\xi=0}^{N-1} \left[ \frac{2\pi |N-M|}{NM} \xi \left| \sum_{n=1}^{N} a(x_n) n e^{-i2\pi \frac{n}{N}\xi} \right| + \left| \sum_{n=N+1}^{M} a(x_n) e^{-i2\pi \frac{n}{M}\xi} \right| \right].$$

(9)

*where both* $C$ *and* $C_\ell$ *are constants independent of* $N$, $M$, *and* $L$.

*Proof.* We aim to establish the bound equation 9 on the discretization mismatch error $E_{MN}$ by extending Lemma B.1 to the composite mapping $\mathcal{G} = Q \circ \mathcal{L} \circ P$ with $d_c$ hidden channels. Without the loss of generality, we assume $s = \min\{s, s'\}$.

Let $a_N = a|_{\Omega_N}$ and $a_M = a|_{\Omega_M}$ denote the discretizations of the input function $a$.

Applying the lifting layer $P$ to $a_N$ and $a_M$, we obtain

$$a_N^0 = P(a_N), \quad a_M^0 = P(a_M),$$

where the superscript denotes the number of Fourier layers that the latent feature functions have gone through.

By the linearity of $P$, $Q$, and the channel mixing of intermediate layers, and $a \in H^s(\mathbb{T}^d; \mathbb{R}^{d_a})$, we have:

$$E_{NM} = \left\| \mathcal{G}\left(a|_{\Omega_N}\right) - \mathcal{G}\left(a|_{\Omega_M}\right) \right\|_{L^2(\mathbb{T}^d; \mathbb{R}^{d_a})}$$
$$\leq \|Q\| \cdot \|P\| \cdot \left\| \mathcal{L}\left(a_N^0\right) - \mathcal{L}\left(a_M^0\right) \right\|_{L^2(\mathbb{T})}$$
$$= C \cdot \left\| \mathcal{L}'(a) - \mathcal{L}'(a) \right\|_{L^2(\mathbb{T}^d; \mathbb{R}^{d_c})},$$

(10)

where $C$ is a constant depends on all the linear channel mixings and independent of $N$, $M$, and $L$, and $\mathcal{L}'$ is the induced single channel operator of Fourier layers.

To estimate the error propagation through $\mathcal{L}$, we utilize Lemma B.1, which provides an error bound for a single-channel case on $\mathbb{T}$. We extend this result to the multi-channel, $d_c$-dimensional setting by considering each channel separately and summing the contributions. The error in each channel propagates similarly due to the Lipschitz continuity of $\sigma$ and the boundedness of the operators $\mathcal{W}_\ell$ and $\mathcal{K}_\ell$. We can bound the error after $L$ layers by:

$$E_{NM} = \|\mathcal{G}\left(a|_{\Omega_N}\right) - \mathcal{G}\left(a|_{\Omega_M}\right)\|_{L^2(\mathbb{T}^d)}$$

$$\leq C\omega^L \left(\prod_{\ell=1}^{L} C_\ell\right) \sum_{\xi=0}^{N-1} \left[\frac{2\pi|N-M|}{NM}\xi \left|\sum_{n=1}^{N} a(x_n)ne^{-i2\pi\frac{n}{N}\xi}\right| + \left|\sum_{n=N+1}^{M} a(x_n)e^{-i2\pi\frac{n}{M}\xi}\right|\right].$$

$$(11)$$

This completes the proof.

$\square$

Now, Proposition 4.4 follows immediately. The interpretation of this bound is similar to that of Lemma B.1. Note that we can fix $N$ or $M$ to be the training resolution; thus, this bound suggests that as long as the inference resolution deviates from that of the training, DMEs arise. In the main text, we use plain language to ensure easier interpretation for the readers.

## C  FULLY CONNECTED POINT-WISE NETWORK AND BAND-LIMITS

For the definition of band-limited function spaces, we refer readers to Raonic et al. (2023). Once the latent feature function is obtained, a fully connected point-wise network (implemented as a $1 \times 1$ convolution with non-linear activations) is applied to capture information beyond the band-limit.

*Remark* 4.5. Let $v(x) \in \mathcal{B}_w\left(\mathbb{T}^d; \mathbb{R}^{d_L}\right)$ be a band-limited function, and let an analytic non-linear function $\sigma(\cdot)$ be the fixed activation function in the fully connected point-wise network. Then the resulting function may not be band-limited anymore. Additionally, the high frequencies of the resulting function is determined by both the fully connected point-wise network and the learned latent feature function $v(x)$.

*Proof.* As point-wise linear transforms will only result in a linear transform of the feature function. Thus, it does not introduce any high frequency information nor does it involve any interaction between different frequency modes. Without loss of generality, consider only one layer of fully connected point-wise network, and we analyze the non-linear activations. Let $\hat{v}(\xi) = \mathcal{F}[v(x)](\xi)$ denote the Fourier transform of $v(x)$, defined by

$$\hat{v}(\xi) = \int_\Omega v(x)e^{-2\pi i\xi x} \, dx.$$

If $\sigma(x)$ is a smooth function of $x$, we can expand $\sigma$ as follows:

$$\sigma(x) = \sigma(0) + \sigma'(0)x + \frac{1}{2!}\sigma''(0)x^2 + \frac{1}{3!}\sigma'''(0)x^3 + \dots.$$

Thus, we can express $\sigma \circ v(x)$ as:

$$\sigma \circ v(x) = \sigma(0) + \sigma'(0)v(x) + \frac{1}{2!}\sigma''(0)v(x)^2 + \dots.$$

Now, performing the Fourier transformation, we obtain:

$$\mathcal{F}[\sigma \circ v(x)](\xi) = 2\pi\sigma(0)v(\xi) + \sigma'(0)\hat{v}(\xi) + \frac{1}{2!}\sigma''(0)\int_\Omega \frac{d\xi_0}{2\pi}\hat{v}(\xi_0)\hat{v}(\xi - \xi_0)\,d\xi_0$$

$$+ \frac{1}{3!}\sigma'''(0)\iint_\Omega \frac{d\xi_0}{2\pi}\frac{d\xi_1}{2\pi}\hat{v}(\xi_0)\hat{v}(\xi_1 - \xi_0)\hat{v}(\xi - \xi_1)\,d\xi_0 d\xi_1 + \dots.$$

We observe that, without loss of generality, if there exists $n > 1$ such that $\sigma^{(n)}(0) \neq 0$, $\sigma \circ v(x)$ may introduce frequencies beyond the band-limit of $v(x)$.        □

It is important to note that, based on the formulation, the high-frequency components beyond the band-limit are determined by both the fully connected point-wise network and the latent feature function $v$ learned by the intermediate neural operator. The intermediate neural operator is tasked to learn an appropriate latent feature function to capture high-frequency components together with the fully connected point-wise network. Whether or not this structure can be proven to have universality is of interest to future exploration.

### C.1 SPECTRAL COMPARISON: FNO AND CROP

In Figure 4, we present a spectral comparison between FNO and CROP on the Burgers example, adapted directly from Li et al. (2021). Both FNO and CROP demonstrate the ability to learn high-frequency information beyond their truncation or band-limit when performing inference at the same resolution as training. However, under super-resolution inference, high frequencies show a slight deviation from the ground truth. Notably, CROP consistently performs better at higher frequencies in both scenarios. It should be mentioned that this plot is in semilog scale, which visually amplifies the error for high frequencies.

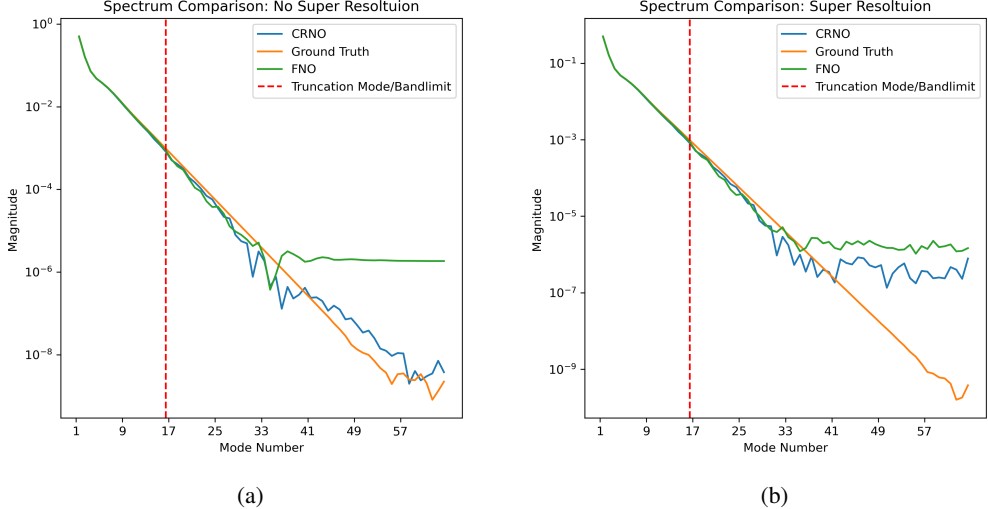

(a)                     (b)

Figure 4: (a): Illustration of the magnitude spectra of CRNO and FNO for inference at the same resolution as training. (b): Illustration of the magnitude spectra of CRNO and FNO for super-resolution inference.

Most importantly, the issue with DMEs is not just on the ability to recover unseen high frequencies. Instead, DMEs cause significant errors in the low-frequency range. We demonstrate this with the spectral error plot for low frequencies, shown in Fig. 5, for the results in Table 5.1.2 (the Navier-Stokes equation). It is evident that DMEs cause high errors in low frequencies. At the training resolution, both FNO and CROP exhibit minimal errors in low frequencies; however, FNO introduces substantial errors in low frequencies during super-resolution inferences, whereas CROP remains consistent.

## D EXPERIMENTAL DETAILS

All the experimental results, especially the timing results, are recorded on NVIDIA RTX A6000 with 48 GB GDDR6.

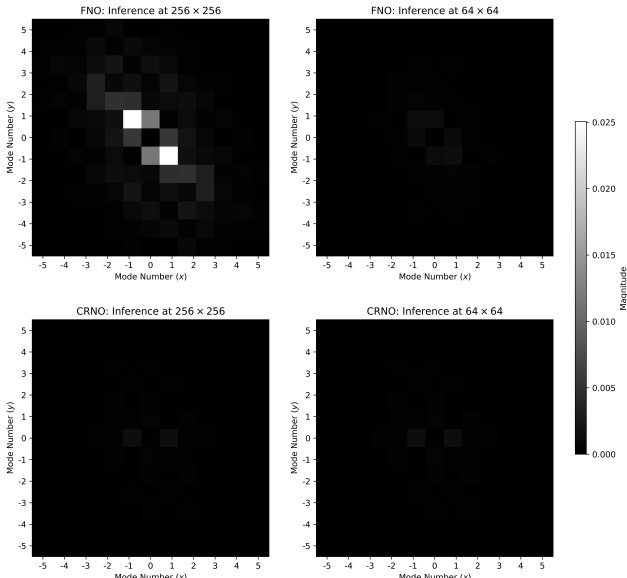

Figure 5: Illustration of the magnitude spectra of the error for FNO and CRNO at a training resolution of $64 \times 64$. It is evident that DMEs cause significant inaccuracies in the low-frequency range.

## D.1 THE NAVIER-STOKES EQUATION

We consider the 2D Navier-Stokes equation for a viscous, incompressible fluid in vorticity form from Li et al. (2021):

$$
\begin{aligned}
\partial_t w(x,t) + u(x,t) \cdot \nabla w(x,t) &= \nu \Delta w(x,t) + f(x), & x \in (0,1)^2, t \in (0,T] \\
\nabla \cdot u(x,t) &= 0, & x \in (0,1)^2, t \in [0,T] \\
w(x,0) &= w_0(x), & x \in (0,1)^2
\end{aligned}
\tag{12}
$$

where $u \in C\left([0,T]; H_{\mathrm{per}}^r\left((0,1)^2; \mathbb{R}^2\right)\right)$ for any $r > 0$ is the velocity field, $w = \nabla \times u$ is the vorticity, $w_0 \in L_{\mathrm{per}}^2\left((0,1)^2; \mathbb{R}\right)$ is the initial vorticity, $\nu$ is the viscosity coefficient, and $f \in L_{\mathrm{per}}^2\left((0,1)^2; \mathbb{R}\right)$ is the forcing function.

### D.1.1 THE ACCUMULATION OF DISCRETIZATION ERRORS IN SEC. 5.1

For Sec. 5.1, $\nu$ is set to $1\mathrm{e}-3$. We are interested in learning the operator mapping the vorticity up to time 10 to the vorticity up to a later time $T = 50$ in an auto-regressive manner for all the intermediate time steps with a step size of 1 as in Li et al. (2021). This dataset is generated using the data generation scripts provided by the authors of Li et al. (2021) in their GitHub repository. The PDE is solved using numerical solvers on a $256 \times 256$ grid. The training dataset contains 1,000 trajectories (input-output function pairs), while the testing dataset includes 200 trajectories. Following the setups in Li et al. (2021), we downsample the training dataset to $64 \times 64$; the FNO model is fully trained on this data and evaluated on the testing data at different resolutions, namely $256 \times 256$, $128 \times 128$, $64 \times 64$, and $32 \times 32$, also obtained through downsampling.

For Figure 3 (a), the FNO architecture is followed exactly as provided by the authors of Li et al. (2021) in their GitHub repository (4 Fourier Layers and a hidden channel dimension of 20) except that the teacher forcing training strategy from Tran et al. (2023) is adopted. The FNO model is trained on $64 \times 64$ data and we make inferences on different resolutions mentioned above.

The relative $\ell_2$ error is given by

$$
\text{relative } \ell_2 \text{ error} = \frac{\|u_\theta - u\|_2}{\|u\|_2},
$$

where $u_\theta$ is the prediction and $u$ is the label produced by numerical solvers. Since the obtained results are discrete, the 2-norm is calculated as the 2-norm of vectors. We report the relative $\ell_2$ error for all results in this paper unless otherwise specified. The experiment is repeated 10 times, and we record the mean.

In Figure 3 (b), we consider the same Navier-Stokes Equation and generated data. Due to the tremendous computational resources it might take, we are only interested in learning the mapping $G : u_{10} \mapsto u_{11}$ without auto-regressive time steps, where $u_t$ denotes the vorticity at time $t$. The FNO is trained on data at $64 \times 64$, and we compare the difference between the FNO output when taking input data of resolution $64 \times 64$ and that of $256 \times 256$. The relative DME is given by

$$\frac{\left\| \mathcal{G}_\theta \left( u_{10}|_{\Omega_{64}} \right) - \mathcal{G}_\theta \left( u_{10}|_{\Omega_{256}} \right) \right\|_2}{\|u_{11}\|_2},$$

where the numerator is the DME and the denominator $\|u_{11}\|_2$ is applied to help the readers to compare the DME relative to the label solution to which the neural operator is trained to map. As $\mathcal{G}_\theta \left( a|_{\Omega_{64}} \right)$ and $\mathcal{G}_\theta \left( a|_{\Omega_{256}} \right)$ practically have different resolutions, the 2-norm is calculated using the Parsevals identity: $\|v\|_2 := \frac{(2\pi)^d}{2} \sum_{k \in \mathbb{Z}^d} |\widehat{v}_k|^2$ for a mapping $v : \mathbb{T}^d \mapsto \mathbb{R}^d$, where $\widehat{v}_k = \mathcal{F}(v)(k)$ denotes the $k$-th Fourier coefficient of $v$ for any $k \in \mathbb{Z}^d$.

The experiment is repeated 20 times; the mean and standard deviation are reported in Table 5. In addition, we observe that for different datasets and different settings of FNO, the DME might fluctuate or not increase much after a few layers (potentially due to residual connections). As we are interested in the upper bound, we report one set of results that clearly reflects our findings. However, for all the experiments we have conducted, DME clearly exists, even with only 2 to 3 Fourier layers.

Table 5: The growth in relative discretization mismatch error (in percentage) with respect to increasing numbers of layers. This highlights our observation that discretization mismatch errors propagate through Fourier layers.

| Number of Layers | 1 | 2 | 3 | 4 | 5 |
|---|---|---|---|---|---|
| Mean (Std) | 0.28($\pm$0.09) | 0.45($\pm$0.09) | 0.43($\pm$0.07) | 0.46($\pm$0.11) | 0.43($\pm$0.13) |
| Number of Layers | 6 | 7 | 8 | 9 | 10 |
| Mean (Std) | 0.46($\pm$0.12) | 0.52($\pm$0.13) | 0.65($\pm$0.26) | 0.91($\pm$0.45) | 1.55($\pm$0.69) |
| Number of Layers | 11 | 12 | 13 | 14 | 15 |
| Mean (Std) | 1.90($\pm$0.76) | 2.73($\pm$0.77) | 3.30($\pm$0.88) | 3.93($\pm$0.70) | 4.19($\pm$0.62) |

### D.1.2  CROSS RESOLUTION TEST IN SEC. 5.1

For the cross-resolution test of different models in Sec. 5.1, the experiment is repeated 10 times and the mean and standard deviation of the average relative $\ell_2$ error over 40 auto-regressive time steps are plotted and also given in Table 8 in percentage.

We implemented two versions of the alias-free activation functions. For the $2\times$ upsampling implementation, which is directly taken from Raonic et al. (2023), we increase the spatial grid size by a factor of two by bicubic spline interpolation, e.g. $64 \times 64$ to $128 \times 128$, apply the activation functions on higher-resolution grids, then downsample to the original grid size by, again, bicubic spline interpolation. For the fixed upsampling implementation, inputs of different grid sizes are all interpolated to the same size (e.g. $128 \times 128$) to apply the activation functions, and then interpolated back to their original grid sizes. The main difference between these two is that, during inference, the grid sizes also increase by a factor of two in the $2\times$ upsampling implementation while they are fixed in the fixed upsampling implementation.

### D.1.3  THE NAVIER-STOKES EQUATION WITH HIGH REYNOLDS NUMBERS

In Sec. 5.2.1, $\nu$ is set to $5\mathrm{e}-4$ and $1\mathrm{e}-4$, respectively for Reynolds numbers $5,000$ and $10,000$. We are interested in learning the operator mapping the vorticity from time $0$ to the vorticity at time $T = 10$ and $T = 5$ for Reynolds numbers $5,000$ and $10,000$, respectively. This dataset

Table 6: Average $\ell_2$ error during inference for the Navier-Stokes Equation under different resolutions of input parametric functions. CROP result with more significant digits. We can rarely observe any difference between inferences at different resolutions.

|  | $256 \times 256$ | $128 \times 128$ | $64 \times 64^{\diamond}$ | $32 \times 32$ |
|---|---|---|---|---|
| CROP | $0.54137(\pm 0.0529)$ | $0.54140(\pm 0.0527)$ | $0.54154(\pm 0.0527)$ | $0.54142(\pm 0.0526)$ |

$^{\diamond}$ Training resolution is $64 \times 64$.

is generated using the data generation scripts provided by the authors of Li et al. (2021) in their GitHub repository. The PDE is solved using numerical solvers on a $256 \times 256$ grid. For Reynolds numbers $5,000$, the dataset contains $1,024$ trajectories and follows a $768/128/128$ split. As higher Reynolds number leads to a more difficult learning task, for Reynolds numbers $10,000$, the dataset contains $2,048$ trajectories and follows a $1792/128/128$ split.

## D.2 THE DARCY FLOW EQUATION

We consider the steady-state of the 2D Darcy Flow equation given by:

$$-\nabla \cdot (a(x)\nabla u(x)) = f(x) \quad x \in (0,1)^2$$
$$u(x) = 0 \qquad x \in \partial(0,1)^2$$

where $a(x)$ is the coefficient function and $f(x)$ is the forcing function. We learn the non-linear operator mapping $a \mapsto u$ following the exact same setup and data as in Li et al. (2021) and the operator mapping $f \mapsto u$ following the generation setup in Hasani & Ward (2024), where the diffusion coefficient, $a(x)$, that accounts for anisotropy in the flow is taken as

$$a(x) = \begin{bmatrix} x_1^2 & \sin(x_1 x_2) \\ x_1 + x_2 & x_2 \end{bmatrix},$$

where $x_1$ and $x_2$ are the two dimensions of $x$. We are interested in learning the solution operator, $G : f \mapsto u$, mapping the source term $f(x)$ to the solution $u(x)$. 1000 input-output pairs of resolution $256 \times 256$ are generated based on Hasani & Ward (2024) and the program provided in their GitHub repository. The data is downsampled to $64 \times 64$ for training, and the train/val/test data follows a $800/100/100$ split.

**Cross-Resolution Results on the Non-linear Mapping $a \mapsto u$.** In Table 7, we present the comparison of cross-resolution tasks between CROP and FNO. Clearly, CROP remains consistent under different resolutions of inference, while FNO suffers from DMEs. This is not an autoregressive task, so the DMEs do not propagate through time; however, we still observe a significant downgrade in performance for FNO.

Table 7: The mean and standard deviation of relative $\ell_2$ error over 10 runs during inference are reported for the nonlinear operator $a \mapsto u$ on the Darcy flow equation with varying resolutions of input parametric functions. CROP is consistent across all resolutions. FNO shows a strong bias toward the training resolution. The DME is obviously shown when inferred on data with different resolutions.

|  | $421 \times 421$ | $141 \times 141$ | $85 \times 85^{\diamond}$ | $43 \times 43$ |
|---|---|---|---|---|
| FNO | $0.87(\pm 0.04)$ | $0.75(\pm 0.03)$ | $0.68(\pm 0.03)$ | $0.97(\pm 0.03)$ |
| CROP (ours) | $\mathbf{0.52(\pm 0.01)}$ | $\mathbf{0.51(\pm 0.01)}$ | $\mathbf{0.51(\pm 0.01)}$ | $\mathbf{0.57(\pm 0.01)}$ |

$^{\diamond}$ Training resolution is $85 \times 85$.

## D.3 THE POISSON EQUATION

We consider a prototypical 2D Poisson equation with a classical Dirichlet boundary condition from Raonic et al. (2023) given by

$$-\Delta u(x) = f(x), \quad x \in (0,1)^2$$
$$u(x), \quad x \in \partial(0,1)^2 \tag{13}$$

The solution operator, $G : f \mapsto u$, maps the source term $f(x)$ to the solution $u(x)$. The source term is given by

$$f(x, y) = \frac{\pi}{K^2} \sum_{i,j=1}^{K} a_{ij} \cdot \left(i^2 + j^2\right)^{-r} \sin(\pi i x) \sin(\pi j y), \quad \forall (x, y) \in D,$$

with $r = -0.5$, the corresponding exact solution can be analytically computed by

$$u(x, y) = \frac{1}{\pi K^2} \sum_{i,j}^{K} a_{ij} \cdot \left(i^2 + j^2\right)^{l-1} \sin(\pi i x) \sin(\pi j y).$$

The data is generated by fixing $K = 16$ and choosing $a_{ij}$ to be i.i.d. uniformly distributed from $[-1, 1]$. We directly use the data provided by Raonic et al. (2023) and follow their setups.

### D.4 THE HELMHOLTZ EQUATION

We consider the 2D Helmholtz equation from De Hoop et al. (2022) given by

$$\begin{aligned}
\left(-\Delta - \frac{\omega^2}{c^2(x)}\right) u &= 0 && \text{in } \Omega, \\
\frac{\partial u}{\partial n} &= 0 && \text{on } \partial\Omega_1, \partial\Omega_2, \partial\Omega_4, \\
\frac{\partial u}{\partial n} &= u_N && \text{on } \partial\Omega_3,
\end{aligned} \tag{14}$$

where $\Omega = [0, 1]^2$ and $\{\Omega_i\}_{i=1}^4$ are the four edges of the square domain. $\omega$ is set to $10^3$, $c : \Omega \to \mathbb{R}$ is the wave-speed field, and $u : \Omega \to \mathbb{R}$ is the excitation field that solves the equations. The Neumann boundary condition imposed on $\partial\Omega$ is non-zero only on the top edge $\Omega_3$. Throughout the experiments presented in this work, $u_N$ is fixed at $1_{\{0.35 \le x \le 0.65\}}$ following De Hoop et al. (2022). The wave-speed field $c(x)$ is assumed to be

$$c(x) = 20 + \tanh(\tilde{c}(x)),$$

where $\tilde{c}$ is a centered Gaussian

$$\tilde{c} \sim \mathbb{N}(0, C) \quad \text{and} \quad C = \left(-\Delta + \tau^2\right)^{-d}.$$

Here $-\Delta$ denotes the Laplacian on $D_u$ subject to homogeneous Neumann boundary conditions on the space of spatial-mean zero functions with $d = 2$ and $\tau = 3$. We are interested in the operator mapping the wave-speed field $c$ to the solution $u$. The dataset is provided by De Hoop et al. (2022) directly, where this PDE is solved using a finite element method on a $100 \times 100$ grid. We take the first 800 data samples as the training data and the next 200 data samples as the testing. Additionally, the data is normalized to $[0, 1]$ based on the training data.

### D.5 ADDITIONAL CROSS-RESOLUTION RESULTS

Table 8: The Relative $\ell_2$ error for varying resolutions of input parametric functions. In the Navier-Stokes example, the $256 \times 256$ data does not contain more high-frequency information. We can clearly see that CRNO performs consistently.

| | Navier-Stokes ($R_e = 10,000$) | | Darcy Flow | |
|---|---|---|---|---|
| | $256 \times 256$ | $64 \times 64^\diamond$ | $256 \times 256$ | $64 \times 64^\diamond$ |
| FNO | 8.903623 | 5.789168 | 3.636376 | 1.281445 |
| CROP | 2.466607 | 2.466613 | 2.804997 | 1.096139 |

We provide the cross-resolution results for FNO and CROP in Table D.5. Other models either cannot directly perform cross-resolution tasks (e.g., DeepONet) or exhibit dramatically poor performance (e.g., CNN-based models), which leads us to omit their comparison.

We test the cross-resolution performance for both models using the first batch of data from the test dataset. In the Navier-Stokes example, the $256 \times 256$ data does not contain more high-frequency information than the $64 \times 64$ data. We can clearly see that CRNO performs consistently during inference at different resolutions. In the Darcy flow (linear operator $f \mapsto u$) example, however, the $256 \times 256$ data does contain more high-frequency information, resulting in a degradation for both models. This again demonstrates our concerns about zero-shot super-resolution tasks, which we will discuss further in Appendix E.

### D.6 IMPLEMENTATION DETAILS

Model details:

- FNO (Li et al., 2021): Adopted from the code provided by the authors in their GitHub repository with no change to their architecture except that we increase the modes to $24$ with a width of $16$.

- CNO (Raonic et al., 2023): Adopted from the code provided by the authors in their GitHub repository with no change to their architecture.

- U-Net: Adopted from the code provided by PDEBench (Takamoto et al., 2024) with no change to their architecture at all.

- ResNet: Adopted from the well-known ResNet50 architecture, we made minor adjustments to adapt it for our function mapping tasks instead of classification tasks. Specifically, we removed the final pooling and linear layers, and we used a padding of 1 and a stride of 1 to ensure that all channels maintain the same spatial size as the input. Finally, we employed a $1 \times 1$ convolutional layer to project the feature maps into the final output.

- DeepONet: Adopted from the code provided by the authors of Raonic et al. (2023) in their GitHub repository, we made some corrections to make it runnable on our end. Specifially, a CNN-based network is used as the branch network and an MLP is used as the trunk network.

- CROP (ours): For the Navier Stokes equation with a Reynold number of $1,000$, we apply the FNO as the intermediate neural operator as the pattern is rather global. For all other experiments, we use a U-Net or FNO with local $3 \times 3$ convolution kernels as the intermediate neural operator to capture fine details. Additionally, we find that for some problems, such as the nonlinear Darcy problem and the Navier Stokes equation with low Reynolds number, applying learnable convolution kernels in the CROP lifting and projection layers is prone to overfitting; band-limited identity kernels (i.e. Fourier projection and interpolation operators) are applied.

All models are trained using the hyperparameters in Table 9, except the ones that the original work provides the hyperparameter settings, such as FNO with the Darch flow and the Navier Stokes equation with low Reynolds number, with early stopping implemented if the validation error does not improve over a specified number of epochs. One exception in which we do not apply early stopping is the nonlinear Darcy example, the Helmholtz example, and the Navier Stokes equation with low Reynolds number; we follow the original setups from Li et al. (2021), and there is no validation set. However, for baseline methods, a grid search over the hyperparameter space is still conducted based on test error, which solely benefits the baselines.

Table 9: Training Hyper-parameters

| Hyper-parameter | Value |
|---|---|
| Learning Rate | $0.001$; $0.0005$ for ResNet; $0.0001$ for DeepONet |
| Weight Decay | $1e-6$ |
| Scheduler Step | 10 |
| Scheduler Gamma | 0.98 |
| Epochs | 1000; 2000 for DeepONet |
| Batch Size | 16 |
| Patience for Early Stopping | 100; 500 for DeepONet |

# E  SUPER-RESOLUTION AND ZERO-SHOT SUPER-RESOLUTION

Zero-shot super-resolution refers to performing super-resolution without prior training on high-resolution target data. For FNO (Li et al., 2021), this concept suggests that the model can be trained on lower-resolution data (e.g., $64 \times 64$) and applied to higher-resolution inference (e.g., $256 \times 256$). However, as demonstrated in this work, FNO fails to achieve zero-shot super-resolution due to discretization mismatch errors. Philosophically, without high-resolution information in the training set, it is questionable whether neural operators can infer unseen high-resolution features without leveraging techniques like transfer learning, physics-informed losses, conservation laws, or Hamiltonian principles. Therefore, we encourage the community to shift focus from super-resolution tests for neural operators toward fundamentally improving their design, incorporating physics and prior knowledge about the operator. Despite these challenges, we believe FNO offers strong potential for data-driven super-resolution tasks.

Super-resolution, on the other hand, typically relies on training a model on a dataset that contains high-resolution target data, learning to predict high-resolution details from lower-resolution inputs (Dong et al., 2016; Lim et al., 2017; Liang et al., 2021). The main difference between zero-shot super-resolution and data-driven super-resolution is that high-resolution images are actually seen by the model during training of the model. Interestingly, neural operators recently have been investigated and applied for such tasks; for example, Wei & Zhang (2023) employs a deep operator learning framework for super-resolution tasks in computer vision, specifically targeting natural image applications, and has achieved notable results.

A key task in physical models is reconstruction with limited sensors, which can be framed as a super-resolution problem. The objective is to infer missing information and enhance resolution or fidelity by leveraging underlying physical principles or learned patterns. Specifically, this task involves reconstructing a high-resolution or fine-scale field $f \in \mathbb{R}^h$ from coarse-grained or limited sensor measurements $s \in \mathbb{R}^l$, where $l < h$, and often $l \ll h$. The goal is to learn a mapping $\mathcal{F} : \mathbb{R}^l \to \mathbb{R}^h$ from data, such that $\mathcal{F}(s) \approx f$. Neural operators extend this approach by mapping from fixed-size vectors to continuous functions, i.e., $\mathcal{F} : \mathbb{R}^l \to \mathcal{U}$ for some appropriate function space $\mathcal{U}$. This task aligns well with the DeepONet paradigm (Lu et al., 2021). Indeed, Neural Implicit Flow (Pan et al., 2023), which differs from but shares similarities with DeepONet, has been proposed in this context. We believe that FNO holds strong potential for such tasks as well, especially for applications with symmetries such as modeling climate dynamics on spherical geometries. Although some innovation may be needed to adapt FNO for these applications, it is a very interesting future work to explore.

