# OpenReview forum: "Discretization-invariance? On the Discretization Mismatch Errors in Neural Operators"
_ICLR.cc/2025/Conference — ICLR 2025 Poster_

### Official Review · Reviewer_WGpV · 2024-10-16

**Soundness:** 3
**Presentation:** 3
**Contribution:** 2
**Rating:** 5
**Confidence:** 4

**Summary:**

This paper emphasizes that the colloquial definition of discretization invariance does not align with the formal definition. They also demonstrate both formally and empirically that popular operator learning architectures (such as FNO) do not have constant performance across resolutions for many problems. The authors propose the CROP method to address these issues with bandlimited lifting and projection operators (as well as some pointwise operators to capture high frequency details), and they show improved performance compared to popular neural operator architectures.

**Strengths:**

The paper is well-written, especially for those who may not be experts in neural operator theory. I appreciate the formal definition of discretization invariance from Kovachki et al., 2021 and the discussion of the differences between this definition and the colloquial definition. The experimental results are also quite compelling and suggest that the proposed CROP method vastly outperforms other methods.

**Weaknesses:**

Although the paper has some strengths, there are still some areas of improvement. My first concern is with the novelty of the insights about discretization invariance. Previous works have included similar discussions, such as [1] (See Figure 1 and Section 2), which uses the term “discretization convergence” to refer to the formal definition by Kovachki et al., 2021. Prior works have also demonstrated empirically that FNO performs well in zero-shot super-resolution tasks on smooth problems. Given these results, it seems to me that the primary contribution of this work is some theoretical analysis of the discretization mismatch error (the Lemmas in the appendix) and the CROP method.

In regards to the proposed CROP method, it is unclear to me whether high frequencies are adequately being captured by the proposed pointwise neural network in the projection layer. For some problems, looking at the L2 error may not be sufficient to ensure that high frequencies are captured. As such, I would recommend the authors include additional evaluation metrics, such as plotting the spectrum of each and seeing how CROP vs. FNO perform on the higher frequencies (for reference, see Figure 1 in [1]).

I also have some concerns about the statement of Proposition 4.4. I understand that it is meant to be written informally for a broad audience. However, I believe some aspects of it are potentially misleading: for instance, it is not necessarily always true that the discretization mismatch error must increase with the higher resolution, right? If the operator learns a constant function, for instance, then this would not strictly increase.

**References:**
1. “Neural operators for accelerating scientific simulations and design” (2024).

**Minor notes and typos:**
1. The authors write that “The FNO architecture is not limited to rectangular domains, periodic functions, or uniform grids.” I think this might be a bit misleading, since typical instantiations of FNO are limited by these constraints: the FFT can only be applied on uniform grids, without Fourier continuation, the output of the FNO is periodic, etc. It would be ideal if the authors can add some context to this claim.
2. Typo in line 234 “aN FNO.”
3. I see that the appendix contains proofs of rigorous statements. I would clearly denote that Proposition 4.4 is informal.

**Questions:**

1. In the introduction, the authors claim that “It is widely believed that training an FNO on one resolution allows inference on another without degrading its performance, since FNO operates and parameterizes the kernel on the Fourier space.” Do you have references for this claim?
2. What is L in the equation in Definition 4.1?
3. In Proposition 4.4, it is stated that the discretization mismatch error increases with M, but M does not seem to appear in Lemma B.2. I may be missing something, but how does Proposition 4.4 follow from Lemma B.2?
4. How much were the hyperparameters tuned for the baselines, particularly in the Navier-Stokes setting?
5. What architectures were tested for the intermediate neural operator in CROP?

---

> ### Author Response · Authors · 2024-11-15
> **Author Rebuttal to Reviewer WGpV: Part I**
>
> Thanks a lot for your valuable comments! we have revised the paper and provided point-to-point responses as below.
>
> > W1.1: Novelty of the insights about discretization invariance
>
> 1. **There may be some misunderstanding. Reference [1] does not mention or identify the root cause of why FNO may struggle with zero-shot super-resolution tasks. We focus on this aspect, find the root cause, and provide a solution.** In fact, [1] contains extensive discussion on the ability of FNO to perform zero-shot super-resolution tasks, such as:
>
> "They can extrapolate and predict solutions at new locations unseen during training, i.e., perform zero-shot super-resolution." (direct quote from the abstract)
>
> "Fig. 1 demonstrates that empirically, FNOs and other Neural Operators can do zero-shot super-resolution and extrapolate to unseen higher frequencies" (direct quote from Sec 2.1 in [1])
>
> "zero-shot super-evaluation, where a trained Neural Operator can be evaluated on a new, finer discretization, than what is seen during training, and produce a solution on that new resolution, thus potentially resolving
> higher frequency details" (direct quote from Sec 2.2 in [1])
>
> 2. **Our contribution extends beyond theoretical analysis**; we offer valuable insights into this issue, **providing systematic studies and a practical solution**. We believe this is a critical problem to address for advancing the development of infinite-dimensional learning paradigms.
>
>
> > W1.2: Prior works have also demonstrated empirically that FNO performs well in zero-shot super-resolution tasks on smooth problems.
>
> 1. **The Navier-Stokes example in Sec. 5.1 represents a relatively smooth problem, where FNO does not exhibit superior performance in cross-resolution tasks.**
>
> 2. If your concern is on whether or not FNO really struggles with zero-shot super resolution tasks, **FNO-based models struggling with super resolution tasks can be found in several recent works, spanning both autoregressive rollout taks and one-to-one mapping tasks.** To name a few:
>
> In [3], FNO has an relative $\ell_2$ error of **$8.41$%** (from their Table 1 row 1) on the Navier Stokes equation with $\nu=1 \mathrm{e}-4$ and an super resolution test error of **$43.02$%** (from their Table 2 row 1).
>
> In [3], an FNO trained on a resolution of $43\times 43 (s=43)$ reported the following relative $\ell_2$ errors (from their Table 3) on Darcy flow equations:
>
> | Resolution (s)      | Reported Error |
> |--------------------|----------------|
> | 61 x 61 (s=61)     | 0.1164         |
> | 85 x 85 (s=85)     | 0.1797         |
> | 141 x 141 (s=141)  | 0.2679         |
> | 211 x 211 (s=211)  | 0.3160         |
> | 421 x 421 (s=421)  | 0.3631         |
>
> In [4], an FNO reported the following relative mean square errors (from their Table 5) on Darcy flow equations:
>
> | Resolution       | Reported Error           |
> |------------------|-------------------|
> | $ \frac{1}{2} \times $ | $ 1.475 \cdot 10^{-1} $ |
> | $ 1 \times $           | $ 5.867 \cdot 10^{-2} $ |
> | $ 2 \times $           | $ 8.646 \cdot 10^{-2} $ |
> | $ 4 \times $           | $ 7.731 \cdot 10^{-2} $ |
>
> In [4], an SFNO reported the following relative mean square errors (from their Table 5) on Spherical Shallow Water Equations:
>
> | Resolution       | Reported Error                      |
> |------------------|----------------------------|
> | $ \frac{1}{2} \times $ | $ 1.342 \cdot 10^{-3} $ |
> | $ 1 \times $           | $ 9.220 \cdot 10^{-4} $ |
> | $ 2 \times $           | $ 3.830 \cdot 10^{-3} $ |
> | $ 4 \times $           | $ 4.419 \cdot 10^{-3} $ |
>
>
> > W2: Spectrum comparison between FNO and CROP
>
> 1. There might be some misunderstanding here. **The limitations of FNO extend beyond the inability to learn unseen high-frequency information; DMEs also introduce errors in the low-frequency components.** Therefore, CROP's primary purpose is not to surpass FNO in generalization to unseen high-frequency information.
>
> 2. For reference, we have included a spectral comparison between FNO and CROP in Appendix C.1 (Fig. 4). **The results show that CROP is at least on par with FNO in terms of learning higher frequencies beyond the band-limit under both same-resolution inference and high-resolution inference.**
>
> 3. **The limitations of FNO extend beyond learning unseen high-frequency information; DMEs introduce large errors in the low-frequency components.** Similarly, we provide the spectral difference plot in Fig. 5 in Appendix C.1, which clearly reveals the superiority of CROP compared to FNO.

---

> ### Author Response · Authors · 2024-11-15
> **Author Rebuttal to Reviewer WGpV: Part II**
>
> > W3 (Q3): statement of Proposition 4.4; omitted M in Lemma B.2.
>
>   1. What we have shown in Lemma B.2 is that, if for training at a fixed finer resolution $M$, **the discretization mismatch error increases as the inference resolution $N$ decreases** (i.e., as the testing resolution gets lower, the upper bound of the DMEs gets larger). This works the other way as well.
>
>
>   2. Basically, as long as the inference resolution differs from that of training, it will suffer from DMEs, whether it is super-resolution or sub-resolution (lower resolution), as demonstrated in the experimental results in Sec. 5.1.
>
>   3. We simplified the proof a little for the general audience, and along the way, this caused some confusion. **We have updated the descriptions in Proposition 4.4 and included the full proof without simplification to avoid any confusion.** The proof is similar to that of the previous one, especially in the treament of the terms $\omega^L$ and $\prod_{\ell=1}^L C_{\ell}$, although additional efforts are included on the Fourier analysis.
>
>   4. We have also added a sentence (line 288) to help readers better understand this. Additionally, we have included interpretations of the upper bound after the proof (lines 820 and 874).
>
> > Minor notes
>
> Thank you, we have taken actions to address all the minor notes.
>
> > Q1: References for the belief in the ability to perform cross-resolution tasks
>
> 1. The zero-shot super-resolution capabilities of FNO are highlighted in the original FNO paper [1], and many recent works have conducted zero-shot super-resolution tests for this reason, such as [2,3,4] (there are a lot more, we just provide a few), although the performance often falls below expectations.
>
> > Q2: What is L in the equation in Definition 4.1?
>
> This should be $x \in \Omega_J$. Thank you for catching this typo.
>
> > Q3: Lemma B.2
>
> Please refer to our answers on W3
>
> > Q4: How much were the hyperparameters tuned for the baselines, particularly in the Navier-Stokes setting?
>
> 1. For the NS example in Sec. 5.1, we use the original hyperparameters from [1]. The original reported error is $1.28\%$, we report much better error ($0.58\%$ for FNO itself) due to the updates made by the authors to FNO after the release of the paper and the teaching forcing training stategy.
>
> 2. For the NS examples in Sec. 5.2.1:
>   - For FNO, we use the original set up from [1], except that we increase the number of modes to $24$ (originally $12$). We have also tried larger number of modes and channel width, we do not observe noticable improvements besides largely increasing the number of parameters; therefore, we use this setting.
>   - For CNO, we use their original set up from [5], we also used their model selection strategy to select the best performing model hyperparameters.
>   - For U-Net and ResNet, we directly use popular implementations without any modification.
>   - For DeepONet, there are different choices for the branch net. We have tried using MLPs, U-Nets, and even FNOs as the branch, however, none of these attempts are very succefully in reducing the errors. Similar performance of DeepONet on NS equations has also been noted in [5] (Table 1).
>
> References:
>
> [1] Neural operators for accelerating scientific simulations and design, Kamyar Azizzadenesheli et al., Nature Review Physics
>
> [2] Group Equivariant Fourier Neural Operators for Partial Differential Equations, Jacob Helwig et al., ICML 2023
>
> [3] Improved Operator Learning by Orthogonal Attention, Zipeng Xiao et al., ICLR 2024
>
> [4] Neural Operators with Localized Integral and Differential Kernels, Miguel Liu-Schiaffini el al., ICML 2024
>
> [5] Convolutional Neural Operators for robust and accurate learning of PDEs,
> Bogdan Raonić, et al., NeurIPS 2023

---

> > ### Author Response · Authors · 2024-11-17
> > **Friendly Reminder: Rebuttal Period for ICLR Submission**
> >
> > Dear Reviewer WGpV,
> >
> > We are writing to kindly remind you that the rebuttal phase for our ICLR submission is ongoing; we posted our responses and paper revision almost 3 days ago. **If you have any additional feedback, concerns, or questions regarding our response, we would greatly appreciate hearing from you.**
> >
> > We have made tremendous efforts to address your concerns. Specifically, we have provided evidence that the work you mentioned has a completely different focus from ours. We have also provided evidence that the super-resolution performance of FNO falls below expectations in other works. Last but not least, we have refined the proof and added more intuitive explanations and interpretations of the theoretical statement. Please feel free to let us know if any clarification or further elaboration is needed.
> >
> > Thank you for your time and effort in reviewing our paper and engaging in the discussion process.
> >
> > Best Regards
> >
> > Authors

---

> > > ### Comment · Reviewer_WGpV · 2024-11-18
> > >
> > > Dear authors,
> > >
> > > I appreciate your efforts in addressing my concerns. I have raised my score to reflect this. In particular, I appreciate the rewording of the Proposition and the additional results comparing the learned high frequency components of FNO vs. CROP. With the clarification regarding the focus of this paper, I understand the authors' point of view better now.
> > >
> > > Just to clarify my initial point: What I wanted to emphasize in my review is that the phenomenon of discretization error has been explored in the past, and there is never a guarantee that performance will hold steady across discretizations. However, the practical utility of a particular neural operator method is sometimes related to its ability to generalize across different discretizations (depending on the application), which is why I assume that recent papers ([2,3,4] cited by the authors above) have empirically tested this ability. However, this is not necessarily an indication that "It is widely believed that training an FNO on one resolution allows inference on another without degrading its performance"; to me, it merely serves as one of several possible benchmark tasks for operator learning.
> > >
> > > Although I believe I better understand the authors' viewpoint now, I still do not find the identification of the DME to be the strongest contribution of the paper; I consider the practical proposed CROP method, particularly its improvement in performance over other baselines, to be the more interesting contribution. However, I still have some questions:
> > > - What architectures were tested for the intermediate neural operator in CROP?
> > > - Under certain choices of the intermediate neural operator architecture, the proposed CROP method may become similar to CNO in many respects. Further, it seems that CNO performs closely compared to CROP on a variety of the problems. Did authors test the DME of CNO compared to CROP on more problems (e.g., the setting of Table 2)? In other words, in what ways does CROP address the root cause of DME better than CNO does?
> > >
> > > I appreciate the authors' time and effort to addressing my comments.

---

> > > > ### Author Response · Authors · 2024-11-20
> > > > **Second Round Reply to Revier WGpV: Part I**
> > > >
> > > > Dear Reviewer WGpV,
> > > >
> > > > Thank you very much for your feedback and for taking the time to reassess our work. We greatly appreciate your additional input and have provided our responses below.
> > > >
> > > > > Discretization error has been explored in the past
> > > >
> > > > 1. **Our work is the first to investigate the root cause: discretization mismatch errors (DMEs) propagate through layers.** Empirical evidence has shown that FNO may lack robustness when making inferences across different discretizations or resolutions. However, while other works demonstrate empirical performance, they do not explore the underlying reasons for this behavior.
> > > >
> > > > 2. **This is precisely why we emphasize in our work that this belief is a misconception within the community.** As we stated in our paper, we completely agree with you that there is never a guarantee. However, we acknowledge that **many people do hold the view that operators possess this ability**, and numerous reviewers often require authors to perform and excel on this task. Our purpose is to encourage the community to shift its focus toward leveraging physics and mathematics to perform (learned) super-resolution tasks, as discussed in Appendix E, rather than relying solely on zero-shot super-resolution with purely network designs.
> > > >
> > > > > More evindence on "It is widely believed that training an FNO on one resolution allows inference on another without degrading its performance"
> > > >
> > > > 1. **Our paper reveals that super-resolution tasks might not be a suitable benchmask task for (grid-based) neural operator as they suffer from DMEs.**
> > > >
> > > > 2. **Some people do believe that neural operators possess zero-shot super resolution ability** although you might view it as just a benchmark task. Even in your referenced paper [1], there are extensive portions of text describing the zero-shot super-resolution capabilities of FNO and its ability to generalize to unseen high-frequency information.
> > > >
> > > > 3. **Due to this belief, many assume that these models can infer on different resolutions or even train across varying resolutions.** However, our analysis demonstrates that DMEs inevitably arise in such cases, degrading super-resolution performance and hurting cross-resolution training. Additionally, we provide further evidence about this belief here:
> > > >
> > > >      - Direct Quote from [2]:  "However, it is important to account for neural networks with discretization invariance properties, allowing training and **evaluation on PDE data of various resolutions**." (under Sec 1.1, contribution #3)
> > > >
> > > >      - Direct Quote from [3]: "In particular, one expects **consistent** and convergent predictions as the testing meshes are refined.“ (in the paragraph below equation (2))
> > > >
> > > >      - Direct Quote from [4]: " The input and output functions in neural operators can be at any resolution or on any mesh, and **the output function can be evaluated at any point in the domain**."
> > > >
> > > > > The identification of the DME to be the strongest contribution
> > > >
> > > > 1. We do understand that different people have different views. **We value the insights offered by the analysis of DMEs into why FNO struggles with zero-shot super-resolution tasks.** We also experimentally verify these insights in Sec. 5.1 (error propogation through layers). This could inspire people to design more advanced architectures with discretization-invariance and learning in infinite-dimensions.

---

> > > > > ### Author Response · Authors · 2024-11-20
> > > > > **Second Round Reply to Revier WGpV: Part II**
> > > > >
> > > > > ### **Additional Questions on CROP:**
> > > > >
> > > > > > What architectures were tested for the intermediate neural operator in CROP?
> > > > >
> > > > > 1. For the Navier Stokes equation with a Reynold number of $1,000$, we apply the FNO as the intermediate neural operator as the pattern is rather global. For all other experiments, we use a U-Net as the intermediate neural operator to capture fine details. This detail is stated in Appendix D.5.
> > > > >
> > > > >
> > > > > > General Comparison between CROP and CNO
> > > > >
> > > > > 1. **CNO does not even address DMEs. This is to say, CNO cannot perform cross-resolution tasks directly but rather rely on downsampling for the input and interpolation for the output for super resolution tasks** (the drawbacks of this are discussed in your reference [1], e.g., the spectral inefficiency as you mentioned).
> > > > > 2. **CROP does not use anti-alising up and down activations, which is computationally expensive,** to mitigate aliasing issues (as you can see in Table 3, CROP is ~ 7.73x faster than CNO during training).
> > > > > 3. **Our framework is more flexible and fundamentally different from the scope of CNO**. Unlike CNO, which employs a fixed architecture, our approach allows for a variety of choices for the intermediate neural operators. Even if we were to select an intermediate operator such as a U-Net, CROP remains distinct from CNO because it does not rely on downsampling and anti-aliasing activations. Also CROP includes unique CROP lifting and projection layers. These layers enable the learning of global patterns and the recovery of some high-frequency information that is otherwise lost due to bandlimiting.
> > > > > 4. **We observe a significant improvement over CNO in all examples.** At high Reynolds numbers, CROP reduces CNO's error by $33\%$ while significantly cutting computational time (CROP is approximately 7.73x faster than CNO during training). As operator learning is already a well-established field, demonstrating improvements of one order of magnitude on commonly used datasets will be challenging, if not impossible, to convey effectively. **At lower Reynolds numbers (see results below), the improvements are even more substantial.** This is because the turbulent characteristics and small-scale features of the flow at high Reynolds numbers magnify CNO's limitations in efficiently capturing multi-spatial patterns, particularly global ones. In contrast, CROP offers greater flexibility in selecting its intermediate operator, enabling it to capture complex patterns more effectively.
> > > > >
> > > > > 5. **Additional differences** can be found in "The advantage of CROP in addressing DME over CNO" below.

---

> > > > > > ### Author Response · Authors · 2024-11-20
> > > > > > **Second Round Reply to Revier WGpV: Part III**
> > > > > >
> > > > > > > Additional Results for CNO on Table 2
> > > > > >
> > > > > > 1. **Yes, we do have the results.** However, CNO performs poorly on this example, especially for cross-resolution tasks due to interpolation errors propogated through time steps, so we did not include it in the paper. We present the results in the table below. Please let us know if you think it is necessary to include this result in the paper; we will include it in Appendix D.1.2.
> > > > > >
> > > > > > Training resolution $64 \times 64$ (over $5$ runs as CNO is very slow, ~14 times slower than FNO on Nvidia RTX A6000, to train under this Navier Stokes autogregressive setting):
> > > > > >
> > > > > > | Neural Operator        | **$256 \times 256$**       | **$128 \times 128$**       | **$64 \times 64$ (training)** | **$32 \times 32$**       |
> > > > > > |------------------------|----------------------------|-----------------------------|---------------------------------|---------------------------|
> > > > > > | FNO                    | $4.65 (\pm 1.47)$          | $3.35 (\pm 0.97)$           | $0.58 (\pm 0.04)$               | $9.45 (\pm 2.92)$        |
> > > > > > | CROP (ours)            | **$0.54 (\pm 0.05)$**      | **$0.54 (\pm 0.05)$**       | **$0.54 (\pm 0.05)$**           | **$0.54 (\pm 0.05)$**    |
> > > > > > | CNO (without interpolation)                    | $73.19(\pm 3.77)$          | $58.63(\pm 3.30)$           |$ 1.96 (\pm 0.13)$              |  $77.23(\pm 7.97)$        |
> > > > > > | CNO (with bicubic interpolation)                    | $4.16 (\pm 1.74)$          | $2.96 (\pm 1.09)$           | $ 1.96 (\pm 0.13)$               | $7.67(\pm 1.69)$        |
> > > > > >
> > > > > >
> > > > > > 2.  **We incorporated the main innovation of CNO, the upsampling and downsampling mechanisms, into FNO for comparison purposes** (see Table 2 in our paper for the results). This modification neither improved performance nor enabled zero-shot super-resolution capabilities.
> > > > > >
> > > > > >
> > > > > >
> > > > > >
> > > > > > > The advantage of CROP in addressing DME over CNO
> > > > > >
> > > > > > 1. **As mentioned, CNO does not address DMEs but instead uses interpolation/downsampling to enable inference at different resolutions.** As demonstrated in the results above, CNO without interpolation exhibits large DMEs.
> > > > > >
> > > > > > 2. CROP's lifting layers can **learn global patterns from the full discretization without downsampling, producing more meaningful intermediate latent functions for the intermediate neural operator**. CROP's projection layers can **recover high-frequency components lost due to bandlimiting** (see the spectral comparison between FNO and CROP in Appendix C.1).
> > > > > >
> > > > > > 3. If we downsample to the same resolution as the training resolution for CNO, there will be no DMEs, as the discretizations are identical. **However, CNO is unable to generalize to high frequencies (as you mentioned) under such settings.** Interpolation errors can also be propogated through the network and time steps as demonstrate in the results above.
> > > > > >
> > > > > >
> > > > > >
> > > > > >
> > > > > >
> > > > > > [1] Neural operators for accelerating scientific simulations and design, Kamyar Azizzadenesheli et al., Nature Review Physics
> > > > > >
> > > > > > [2] Deep Operator Learning Lessens the Curse of Dimensionality for PDEs, Ke Chen el al., TMLR 2024
> > > > > >
> > > > > > [3] Positional Knowledge is All You Need:Position-induced Transformer (PiT) for Operator Learning, Junfeng Chen et al., ICML 2024
> > > > > >
> > > > > > [4] Guaranteed Approximation Bounds for Mixed-Precision Neural Operators, Renbo Tu et al., ICLR 2024

---

> > > > > > > ### Comment · Reviewer_WGpV · 2024-11-21
> > > > > > >
> > > > > > > I thank the authors for their responses. I suggest you include the CNO results in the appendix, as they make the paper more complete. However, the CNO without input/output interpolation is perhaps not necessary to include, as in that case, if I am understanding correctly, the model does not converge to the desired local operator as the resolution increases.
> > > > > > >
> > > > > > > My remaining concerns are about the numerical experiments, specifically on why CROP outperforms the other methods by such a large margin, even at times at the training resolution, which is interesting because I do not immediately see a reason for why CROP should outperform other models like CNO at training resolution (could it be due to the difference in parameter count, for instance?). Also, to remove the possibility that the baselines are not fully tuned, were any pretrained models used from other papers?
> > > > > > >
> > > > > > > Some of my other questions align with those asked by Reviewer w3hk, so I will also closely follow that discussion. Given these concerns, I will maintain my score for now.

---

> ### Author Response · Authors · 2024-11-22
> **Third Round Reply to Revier WGpV: Part I**
>
> We deeply appreciate the reviewer's insightful comments and valuable suggestions. In response, we conducted additional experiments and further revised our paper, as detailed below.
>
> > Inclusion of CNO Cross-resolution results
>
> - Thank you for your advice. We have included the results in Table 2 and included a sentence below this table to describe this.
>
> > Performance of CROP
>
> 1. **CROP allows the ability to have both local and global filters** (global convolution kernels/filters similar to that of FNO and local filters that are fixed in size, small convolution kernels e.g. $3\times 3$) as the intermediate discretizations are fixed (multi-spatial-learning as described in the manuscript).
>
>      - **While CNO or U-Net also allow the learning of local and global features through the downsampling path, the global patterns might not be well-learned since the global features have to be extracted from much lower-dimensional feature maps.** For example, the original resolution could be $64 \times 64$, and in the bottleneck layer, it becomes $8 \times 8$. At this stage, a fixed $3 \times 3$ kernel can learn more "global" patterns, but these global features have already lost some, if not a lot, information due to compression to lower dimensions. Moreover, U-Nets or CNO may suffer slightly from spatial misalignment issues [1].
>
>      - **We see CROP as a general pipeline (what P stands for) for multi-spatio and cross-resolution learning rather than a specific architecture.** One may choose different intermediate operators based on their specific learning task.
>
> 2. We have discussions in the manuscript as to why CROP is important. For example, lines 474-485 and 501-522. **We give some more intuition to this outperformance here.**
>
>      - For the NS equations, **FNO tends to over-smooth fine-scale details, leading to degraded performance.** This can be observed through the degraded performance as the Reynolds number increases. As the intermediate discretizations are fixed, our intermediate operator can actually have local operations, such as local convolution operators, in the intermediate layers. It can be clearly seen that CNO performance improves as the Reynolds number increases (from $1,000$ to $5,000$ to $10,000$), potentially because of the increased amount of local details and small-scale features. It is possible that if we further increase the Reynolds number, CNO could outperform CROP.
>
>     - For the Darcy flow and Poisson's equation, the ability to learn both local and global features may explain why CROP outperforms other models. Particularly, for the Poisson equation, all models of a local nature (i.e. with local small convolution kernels) outperform FNO, demonstrating the necessity of local learning features in FNO.
>
>     - In fact, under the same resolution, you might also achieve better performance by incorporating some local components into FNO or some global components into CNO. However, **CROP enables this while maintaining the cross-resolution and discretization convergence (as you mentioned) capability.**
>
> 3. Given these, our CROP outperforms other baselines due to its ability to capture both global and local patterns effectively, as demonstrated in experiments in our paper.

---

> ### Author Response · Authors · 2024-11-22
> **Third Round Reply to Revier WGpV: Part II**
>
> > Fair Comparisons
>
> 1. While we did not use any pretrained models as there aren't pretrained models on the examples we present, we conducted an additional experiment on the non-linear Darcy flow operator. For this particular problem, we used the **exact same data** as in the FNO paper. **In the original paper, they reported an error of $1.08\%$ (Table 4 in FNO paper); later on, the FNO authors updated their architecture, and we used their *updated architecture* and obtained an error of $0.68\%$ (which is better than $1.08\%$). If we were to use the originally reported error or trained model, worse performance would be reported for FNO.** For CNO, we used their architecture directly and performed model selection the authors provided in their git repo. Using their original released architecture, which yielded the best performance during model selection, we obtained an error of $0.59\%$. For other models, we used implementations directly from their respective works or popular implementations, as detailed in Appendix D.5. Additionally, refer to Table 1 from [2]; they conducted experiments on exactly the same data from the FNO paper. For your convenience, we summarize the results in Table 1 from [2]:
>
>     - They report $1.08\%$ for FNO, which is from the original paper. We report much better results for the FNO baseline. **Additionally, the lowest error achieved in [2] on this dataset is $0.69\%$, whereas we reported better results than this for both CNO ($0.59\%$; they did not compare with CNO in their paper) and CROP ($0.51\%$).**
>     - Similarly, for the NS equation with Reynolds number $1,000$, we have also used the updated FNO architecture. The originally reported number is $1.28\%$ (Table 1 in FNO paper), while we reported $0.58\%$ using the same updated FNO architecture (by FNO authors as mentioned above).
>
> 2. We have made efforts to ensure fair comparisons, for example:
>     - For FNO, we tested with different numbers of modes and channel widths. We did not observe a significant increase in performance; in fact, using a very high channel width leads to severe overfitting problems.
>
>    - We used the model selection strategy from CNO (as provided in their repository) and found that their original architecture parameters (with no changes at all) gave the best performance. Some other choices yielded at most similar performance, so we retained the original configuration.
>
> 3. The outperformance of CROP is not attributed to the number of network parameters. We include the following results with parameter counts on the Navier-Stokes equation with a Reynolds number of $10,000$ for CROP. The first configuration is used in Table 3 in the paper. **It is clear that with fewer parameters, the performance does not decrease significantly. Moreover, with much fewer parameters than other baselines, the performance is still better than baseline models.**
>
> | Layers         | Initial Channel Width | Residual Blocks (Middle Layer) | Residual Blocks (Bottleneck Layer) |Error(%)|#Parameters|
> |----|-----|--------|-------|------|-|
> | 3 |  16  |  4|    6     |4.06|4.03|
> | 3 |  8  |  2|    4     |4.24|1.19|
> | 2 |  16  |  2|    6     |4.54|1.32|
> | 2 |  16  |  2|    4     |4.45|1.17|
>
>
> (results are **averaged over three** runs for other configurations in this ablation study)
>
> > Some of my other questions align with those asked by Reviewer w3hk, so I will also closely follow that discussion.
>
> - We have responded to Reviewer w3hk, please refer to the answers in our responses to Reviewer w3hk. In summary, our responses resolve the following concerns:
>
>     - We provided additional explanations to the CROP pipeline; we also explained the different focus between our work and CNO.
>     - We provided high-level ideas on why CROP can reduce the DMEs.
>     - We included additional experiments on the exact same Darcy flow problem from the original FNO paper. We show that CROP can achieve the best performance among all baselines.
>
> References
>
> [1] SineNet: Learning Temporal Dynamics in Time-Dependent Partial Differential Equations, Xuan Zhang et al., ICLR 2024
>
> [2] Improved Operator Learning by Orthogonal Attention, Zipeng Xiao et al.,  ICML 2024

---

> ### Author Response · Authors · 2024-11-25
> **Rebuttal Period Ending Soon**
>
> Dear Reviewer WGpV,
>
> We deeply appreciate your insightful feedback and thoughtful suggestions on our paper throughout the review and rebuttal process. Your input has been invaluable in helping us refine and strengthen our work. After your reply, we have included additional experiments and clarification to resolve your concerns, and we believe that we have resolved reviewer w3hk's concerns as well.
>
> With the discussion period ending soon on **Nov 26, 2024 (Anywhere on Earth)**, we kindly await your feedback or an updated assessment of our paper. Please let us know if your concerns have been satisfactorily addressed; if so, we would greatly appreciate it if you could update your ratings. We remain available and strive to answer any additional questions.
>
>
> Thank you once again, and we wish you all the very best :)
>
> Sincerely,
>
> Authors

---

> > ### Comment · Reviewer_WGpV · 2024-11-26
> >
> > I thank the authors for their time in answering my questions and those of the other reviewers. I think the current state of the paper is much improved. However, due to my initial concerns about the novelty of the theoretical contributions and DME discussion, and after a thorough reading of the discussions above (including those with the other reviewers), I have decided to keep my updated score.

---

> ### Author Response · Authors · 2024-12-02
> **Reply to Reviewer WGpV**
>
> Dear Reviewer WGpV,
>
> We thank Reviewer WGpV for her/his reply and we appreciate her/his opinions, and we are glad to know that she/he thinks the current state of the paper is much improved.
>
> **We have made our efforts to clarify her/his concerns on the novelty of DME discussions, we summarize below:**
>
> 1. There has been a long-standing misconception in the community that neural operators can perform super-resolution tasks without any degradation in performance. We clarify this misconception by finding the root causes leading to such a degradation, Discretization Mismatch Errors(DMEs).
>
> 2. We are the first to identify DMEs (Sec. 4.1), provide theoretical analyses (Sec. 4.2), verify theoretical conclusions with empirical experiments (Sec. 5.1), and propose a solution (Sec. 4.3).
>
> 3. We provided direct quotes from your referenced paper [1] to demonstrate that [1] does not address (or even identify) DMEs in neural operators but rather overlooks their existence and promotes the zero-shot super-resolution capability.
>
> 4. We provided several recent papers to demonstrate that the issue of DMEs exists in other works and significantly impacts performance. Additionally, we presented evidence showing that many in the community are unaware of the issue of DMEs and assume the resolution-invariance properties of neural operators.
>
>
> Given these reasons, we do believe that our work and perspective on DMEs **are novel and provide important insights into the field of neural operators**; however, we completely understand that different people may hold varying views and opinions. We sincerely thank Reviewer WGpV for her/his efforts in helping us improve the paper.
>
>
> [1] Neural operators for accelerating scientific simulations and design, Kamyar Azizzadenesheli et al., Nature Review Physics

---

### Official Review · Reviewer_z3xH · 2024-10-18

**Soundness:** 3
**Presentation:** 4
**Contribution:** 3
**Rating:** 8
**Confidence:** 5

**Summary:**

This paper addresses the challenge of discretization mismatch errors (DMEs) in neural operators, particularly FNOs. The authors identify that while FNOs are theoretically designed to be discretization-invariant, they still exhibit performance inconsistencies across different resolutions due to DMEs. I like that they also provide a thorough mathematical analysis of how these errors accumulate through the layers of the neural network. To address this issue, they propose a novel Cross-Resolution Operator-learning Pipeline (CROP). CROP introduces lifting and projection layers that map input and output functions to and from a band-limited function space, allowing the use of fixed discretization's in the latent space. This approach effectively minimizes DMEs and enables consistent performance across different resolutions. The authors demonstrate CROP's effectiveness through extensive experiments on various partial differential equations, including the Navier-Stokes equation at high Reynolds numbers. Results show that CROP not only achieves robust cross-resolution performance but also outperforms baseline models in learning complex dynamics. Overall, this paper contributes significantly to the field of neural operators by addressing a critical limitation and proposing a solution that maintains the advantages of FNOs while overcoming their resolution-dependent weaknesses. As a person who works primarily on Neural Operators, I figured someday someone would finally do this work: ) and talk about it. I do have certain questions and concerns but overall, i really enjoyed the presentation of this paper.

**Strengths:**

1. Analysis: The paper  identifies and provides a rigorous theoretical analysis of discretization mismatch errors (DMEs) in neural operators, particularly FNOs, addressing a critical gap in the field. The results do make sense to me as it makes sense that although FNO does have the discretization invariance property, just training on low res - isnt guaranteed enough to get the high level features in high -res images.
2. Unique solution: The proposed CROP method effectively mitigates DMEs through a clever use of lifting and projection layers, allowing for consistent cross-resolution performance and flexibility in choosing intermediate neural operators.
3. Strong empirical validation: Extensive experiments across multiple PDEs, including challenging cases like high Reynolds number Navier-Stokes equations, demonstrate CROP's superior performance compared to state-of-the-art baselines.
4. Clear exposition and reproducibility: I like the fact that the paper is well-structured, clearly written, and provides detailed information on experimental setups, enhancing understanding and reproducibility of the results as well as they don't just bash FNO's, they actually justify their reasons well.

**Weaknesses:**

Some weaknesses
1. Variance in some results: Some empirical results, particularly in Figure 3b, show high variance. Maybe some explanation of the high variance?
2. There's minimal discussion on how sensitive CROP is to hyperparameter choices, which is important for understanding its robustness and ease of application to new PDE problems
3. Long-term stability in time-dependent problems: For time-dependent PDEs like Navier-Stokes, there's limited discussion on the long-term stability of CROP predictions over extended time horizons.
4. Limited exploration of high-frequency information: While CROP addresses the issue of DMEs, the paper doesn't deeply explore how effectively it captures high-frequency information, especially in comparison to methods specifically designed for super-resolution tasks.
5. It would be also great to have pseudocode or the full architecture.

**Questions:**

I have several questions so please bare with me:
1  - So, is CROP still a universal approximation for operators? It seems like its not but I may be mistaken.
2 - I do understand the concept of band limiting and then learning a NO in the intermediate representation, but to be more fair in comparison to FNO for high Reynolds tasks, you should use higher number of modes to capture those details or use something like an Incremental Fourier Neural Operator [1]. It would be great to do some ablation studies on that if you have the time.
3 - How sensitive is CROP's performance to the choice of band-limit in the lifting and projection layers? Is there a systematic way to determine optimal band-limits for different types of problems?
4 - How does the computational complexity of CROP compare to standard FNO, especially for very high-resolution inputs?
5 - How does the choice of intermediate neural operator in CROP affect its performance and computational efficiency? Are there certain types of problems where specific architectures are preferable?
6 - Also I do get you have additional experiments on deeper FNO's but it would be great to actually also use skip connections or other techniques to actually train them well. They do suffer from convergence issues.

---

> ### Author Response · Authors · 2024-11-15
> **Author Rebuttal to Reviewer z3xH: Part I**
>
> Thank you so much for your detailed and constructive comments! We appreciate your questions! We have revised the paper and also provide responses here.
>
> > W1: Variance in some results: Some empirical results, particularly in Figure 3b, show high variance. Maybe some explanation of the high variance?
>
> 1. Figure 3b shows the super-resolution errors. **High variance can occur due to the accumulation of errors over sequential steps (layers).** In some cases, the discretization mismatch error in the next step can partially "correct" or offset errors from previous steps, effectively reducing the cumulative error.
>
> 2. Moreover, the use of skip connections might also contribute to this effect. In certain runs, the later layers may simply be learning the identity map when the number of layers is overly sufficient. It is worth investigating further why this occurs and whether it can offer any new insights into designing more robust neural operators.
>
> 3. If you find it necessary to include such discussions in the paper, please let us know, we will include under Appendix D.
>
> > W2: There's minimal discussion on how sensitive CROP is to hyperparameter choices, which is important for understanding its robustness and ease of application to new PDE problems
>
> 1. In Appendix D5, we provide the implementation details. **All training hyperparameters (learning rates etc.) are general for CROP**; we only adjust some of them for the baselines, as we found that they perform better under those settings.
>
> 2. Moreover, the backbone intermediate finite-dimensional operator is a standard U-Net, and **all network parameter choices are standard and commonly used in other works that employ U-Nets** in tasks such as segmentation (depth of 3, initial channel width of 16, kernel size of 3, 4 residual blocks in the middle layers, and 6 residual blocks in the bottleneck layer).
>
> 3. We additionally perform ablation studies on the choices of network parameters on the Navier-Stokes equation with Reynolds number 10,000 (first one is used in the paper; results are averaged over 3 runs for this ablation study). The results reveal that CROP is stable. If you find it neccesary, please let us know, we will include it in Appendix D.5.
>
> | Layers           | Initial Channel Width | Residual Blocks (Middle Layer) | Residual Blocks (Bottleneck Layer) |Error(%)|
> |----|-----|--------|-------|------|
> | 3 |  16  |  4|    6     |4.06|
> | 2 |  16  |  2|    6     |4.54|
> | 2 |  16  |  2|    4     |4.45|
> | 2 |  16  |  4|    6     |3.96|
> | 2 |  32  |  2|    2     |4.77|
> | 2 |  32  |  2|    4     |4.49|
> | 3 |  16  |  2|    4     |4.69|
> | 3 |  8  |  2|    4     |4.24|
> | 3 |  8  |  4|    6     |4.14|
>
> > W3: Long-term stability in time-dependent problems
>
> 1. **As CROP does not suffer from discretization mismatch errors for any time step, the error does not propogate through time.** CROP is stable for long-term roll-outs in terms of cross-resolution abilities. We have included this discussion in the paper (line 442)
>
> 2. As shown in Table 2, CROP maintains consistent results over different resolutions of inference. This example has 40 rollout steps, which demonstates CROP's ability for long-term rollouts under cross-resolution tasks.
>
> 3. If you are referring to the stability of long-time roll-out under the same resolution of inference, this is out of the scope of our work, as specific network and training designs must be implemented (e.g. [3]).
>
> > W4: Limited exploration of high-frequency information: While CROP addresses the issue of DMEs, the paper doesn't deeply explore how effectively it captures high-frequency information, especially in comparison to methods specifically designed for super-resolution tasks.
>
> 1. We have included a spectral comparison between FNO and CROP in Appendix C.1 (Fig. 4). **The results show that CROP is at least on par with FNO in terms of learning higher frequencies beyond the band-limit under both same-resolution inference and high-resolution inference.**
>
> 2. **The limitations of FNO extend beyond learning unseen high-frequency information; DMEs introduce large errors in the low-frequency components.** Similarly, we provide the spectral difference plot in Fig. 5 in Appendix C.1, which clearly reveals the superiority of CROP compared to FNO.

---

> ### Author Response · Authors · 2024-11-15
> **Author Rebuttal to Reviewer z3xH: Part II**
>
> > W5: It would be also great to have pseudocode or the full architecture.
>
> 1. **We have provided a description of the CROP method in Sec. 4.3.**
>
> 2. The framework operates in three steps. First, a CROP lifting layer lifts the input function to higher channel dimensions within bandlimited function spaces to facilitate enriched representation learning. Second, an intermediate neural operator—chosen freely—maps the bandlimited high-dimensional function to another bandlimited high-dimensional function. Third, a CROP projection layer projects the latent function down to the output dimension while attempting to recover some lost high frequencies. If you find it necessary, please let us know, and we will provide more details in the appendix in the paper.
>
> > Q1: So, is CROP still a universal approximation for operators? It seems like its not but I may be mistaken.
>
> 1. **The short answer is YES**, theorectically.
>
> 2. For mappings between bandlimited function spaces, the result is straightforward: the lifting and projection operators serve as mappings within bandlimited function spaces, which can just be identity if needed, and the intermediate neural operator can be chosen to be an operator that possesses universality.
>
> 3. For mappings between general Sobolev spaces, the universality of our approach can be established in a manner similar to that of FNO. For any desired accuracy, a suitable bandlimit can be always be found such that the Fourier projection error remains sufficiently small, allowing us to apply the universality result within bandlimited function spaces. However, as suggested in [1], general infinite-dimensional function spaces such as $H^s$ may be too broad to effectively learn, making bandlimited function spaces of greater interest.
>
> > Q2: use higher number of modes to capture those details
>
> 1. **We actually have already used a larger number of modes for the results presented in the paper**. We use 24 modes as opposed to the original of 12 (we describe the use of 24 modes in implementation details under Sec. D.5); the resolution is 64, so full modes is 33 (in terms of RFFT). For your reference, we also have conducted additional experiments (over 3 runs) with full modes, and it results in an error of $7.27\%$ for Reynolds number $5,000$, and $8.34\%$ for Reynolds number $10,000$ (Similar to that of using 24 modes).
>
> > Q3: Choice of Band-limit
>
> 1. The general strategy is to choose a bandlimit such that the loss of high-frequency information is minimal. However, similar to FNO, there is a trade-off between the bandlimit and computational efficiency. The universality results of FNO are also built upon a sufficient bandlimit (truncation modes) to ensure desired arbitrary small Fourier projection errors.
>
>
> > Q4: How does the computational complexity of CROP compare to standard FNO, especially for very high-resolution inputs?
>
> 1. Overall, it is comparable to that of FNO. The computational complexity of CROP is depends on the band-limit; if the band-limit is kept the same, the computational time will barely increase.
>
> 2. FNO has a complexity of $O(n^2 \log n)$ for 2D examples due to 2D FFT. In general, it's $O(J \log J)$ for a discretization of size $J$ (Assuming we can perform FFT).
>
> 3. Training under high resolutions can be time-consuming to demonstrate; however, we have provided here a comparison of inference times for higher-resolution settings of the Navier-Stokes (NS) equation with a Reynolds number of 10,000. The results below were recorded on an NVIDIA RTX A6000 GPU with 48 GB GDDR6 over a batch of 20 samples. While the training speed of CROP is slower than that of FNO (as seen in Table 3, Sec. 5.2.1), CROP achieves a faster inference time. For high-resolution inputs, CROP is even much faster.
>
> |                       | Navier-Stokes ($R_e = 10,000$) |                     |
> |-----------------------|-------------------------------|---------------------|
> |                       | **256 x 256**                 | **64 x 64**    |
> | **FNO**               | 32.87ms                   | 3.01ms          |
> | **CROP**              | 2.83ms                   | 2.18ms         |

---

> ### Author Response · Authors · 2024-11-15
> **Author Rebuttal to Reviewer z3xH: Part III**
>
> > Q5: Choice of intermediate neural operator in CROP
>
> 1. We found that for problems with multiscale spatial features, such as Navier-Stokes with high Reynolds numbers, an intermediate neural operator with the ability to capture localized structure is generally preferred (we used a U-Net). However, for smooth problems of a global nature, such as Navier-Stokes with a Reynolds number of 1,000 (as shown in Sec. 5.1), a global operator like FNO is more suitable.
>
> 2. In terms of computational efficiency, it largely depends on the choice of intermediate neural operator and the selected bandlimit. However, as demonstrated in the paper, **CROP achieves strong performance while remaining comparable in efficiency to operators like FNO**.
>
> > Q6: Skip connections for FNO.
>
> 1. **In experiments, we do have skip connections.** We use the original implementation provided by [2] (the updated version), and their implementation do have skip connections.
>
> 2. Both training and testing errors at the same resolution as the training data remained fairly low. However, the cross-resolution error increases as the number of layers grows.
>
> References
>
> [1] Convolutional Neural Operators for robust and accurate learning of PDEs,
> Bogdan Raonić, et al., NeurIPS, 2023
>
> [2] Fourier Neural Operator for Parametric Partial Differential Equations, Zongyi Li et al., ICLR 2021
>
> [3] PDE-Refiner: Achieving Accurate Long Rollouts with Neural PDE Solvers, Lippe el al., NeurIPS 2023

---

> > ### Comment · Reviewer_z3xH · 2024-11-17
> > **Response**
> >
> > I thank the authors for their detailed responses and answers to my questions. As of now, I have no questions: )! The revised paper looks great! I will keep my score however.

---

> > > ### Author Response · Authors · 2024-11-17
> > > **Reply by the Authors**
> > >
> > > Thank you for your positive feedback and for taking the time to review our revisions. We appreciate your thoughtful engagement throughout the process. We're glad to hear that the revised paper meets your expectations. Thank you so much again for your time and effort in providing valuable insights to improve our work.

---

### Official Review · Reviewer_FHqG · 2024-10-29

**Soundness:** 4
**Presentation:** 4
**Contribution:** 3
**Rating:** 8
**Confidence:** 4

**Summary:**

This paper first pointed out a common misconception that FNO does not depend on the resolution of input. The authors proved that resolution DOES affect FNO and defined a metric, discretization mismatch error (DME), to quantify the effect of resolution changing. Then they proposed a solution to mitigate the issue that ensures FNO's high performance across different resolutions.

The author provided estimation of DME by mathematical analysis.
Lemma B.1. provides the estimation of difference between input functions of FNO layers at different resolution $N$ and $M$, which is bounded by $o(\frac{1}{N^{s-1}})$ assuming $N<M$ and $s\geq2$. Lemma B.2. extended the analysis of Lemma B.1. to neural operators with lifting and projection layers.

They proposed a solution, cross-resolution operating learning pipeline (CROP), which relies on 1x1 convolution as lifting and projection layers.
Then the authors conducted experiments to show two superior aspects of CROP: cross-resolution tasks and learning capability.

**Strengths:**

**Originality** Though the development of neural operators has been accelerating recently, some fundamental concepts need clarification. I really appreciate the contribution of this paper that addresses the particular misconception of cross-resolution ability of FNO and puts neural operators on solid ground. The solution proposed here is based on band-limited function spaces and implemented as 1x1 convolution.

**Quality** The mathematical analysis is solid and experiment is persuasive.

**Clarity** I find the paper clear to read and carefully composed.

**Significance** is relatively high. Cross-resolution application is a favorable feature of neural operators like FNO. The price to pay for such feature should be reminded for researchers in this field.

**Weaknesses:**

Since your work is to some degree a direct challenge to part of the arguments in [1], can you provide some insights to their evidence of "discretization invariance" that contradicts your findings? Namely, in Section 7 of [1], several experiments support the argument "the error of FNO is independent of resolution or any specific discretization", see Fig. 8 and Table 2&3. How would you explain such results?


[1] Nikola Kovachki, Zongyi Li, Burigede Liu, Kamyar Azizzadenesheli, Kaushik Bhattacharya, Andrew
Stuart, and Anima Anandkumar. Neural operator: Learning maps between function spaces
with applications to pdes. Journal of Machine Learning Research, 24(89):1–97, 2023.

**Questions:**

N.A.

---

> ### Author Response · Authors · 2024-11-15
> **Author Rebuttal to Reviewer FHqG**
>
> Thank you so much for your efforts reviewing our work. We respond to your questions below.
>
> > W1.1: Contradiction between [1] and our findings
>
> 1. There might be a misunderstanding here; **our work does not challenge [1]** but rather the terminology used FNO is indeed discretization-invariant as defined in [1] (or Definition 4.2 in our paper). However, **this definition does not enable FNOs to perform zero-shot super-resolution tasks (or generalize well to unseen high frequencies)**. By this definition, even U-Net can be considered discretization-invariant, as its local convolutions converge to a point-wise operator. Therefore, we propose the term 'convergence' for this definition.
>
> 2. In summary, our work does not challenge [1], but rather **clarify a misconception in the community.** We discuss this throughout Sec. 4.1, e.g. lines 256-268.
>
> > W1.2: Section 7,ig. 8 and Table 2&3 of [1] Fig. 8 and Table 2&3.
>
> 1. For the results in Fig. 8 and Table 2&3 in [1], **the training and testing resolutions are the same** (the caption of Fig. 8: "Train and test on the same resolution"), which is not what it means to perform super-resolution (cross-resolution) tasks.
>
> 2. If your concern is on whether or not FNO really struggles with zero-shot super resolution tasks, **FNO-based models struggling with super resolution tasks can be found in several recent works, spanning both autoregressive rollout taks and one-to-one mapping tasks.** To name a few:
>
> In [3], FNO has an relative $\ell_2$ error of **$8.41$%** (from their Table 1 row 1) on the Navier Stokes equation with $\nu=1 \mathrm{e}-4$ and an super resolution test error of **$43.02$%** (from their Table 2 row 1).
>
> In [3], an FNO trained on a resolution of $43\times 43 (s=43)$ reported the following relative $\ell_2$ errors (from their Table 3) on Darcy flow equations:
>
> | Resolution (s)      | Reported Error |
> |--------------------|----------------|
> | 61 x 61 (s=61)     | 0.1164         |
> | 85 x 85 (s=85)     | 0.1797         |
> | 141 x 141 (s=141)  | 0.2679         |
> | 211 x 211 (s=211)  | 0.3160         |
> | 421 x 421 (s=421)  | 0.3631         |
>
> In [4], an FNO reported the following relative mean square errors (from their Table 5) on Darcy flow equations:
>
> | Resolution       | Reported Error           |
> |------------------|-------------------|
> | $ \frac{1}{2} \times $ | $ 1.475 \cdot 10^{-1} $ |
> | $ 1 \times $           | $ 5.867 \cdot 10^{-2} $ |
> | $ 2 \times $           | $ 8.646 \cdot 10^{-2} $ |
> | $ 4 \times $           | $ 7.731 \cdot 10^{-2} $ |
>
> In [4], an SFNO reported the following relative mean square errors (from their Table 5) on Spherical Shallow Water Equations:
>
> | Resolution       | Reported Error                      |
> |------------------|----------------------------|
> | $ \frac{1}{2} \times $ | $ 1.342 \cdot 10^{-3} $ |
> | $ 1 \times $           | $ 9.220 \cdot 10^{-4} $ |
> | $ 2 \times $           | $ 3.830 \cdot 10^{-3} $ |
> | $ 4 \times $           | $ 4.419 \cdot 10^{-3} $ |
>
> References:
>
> [1] Neural operator: Learning maps between function spaces with applications to PDEs. Nikola Kovachki et al., JMLR 2023
>
> [2] Group Equivariant Fourier Neural Operators for Partial Differential Equations, Jacob Helwig et al., ICML 2023
>
> [3] Improved Operator Learning by Orthogonal Attention, Zipeng Xiao et al., ICLR 2024
>
> [4] Neural Operators with Localized Integral and Differential Kernels, Miguel Liu-Schiaffini el al., ICML 2024

---

> ### Author Response · Authors · 2024-11-25
> **Rebuttal Period Ending Soon**
>
> Dear Reviewer FHqG,
>
> We greatly appreciate your thoughtful comments on our paper. We kindly remind you that the discussion period ends soon on **Nov 26, 2024 (Anywhere on Earth)**. If we have adequately addressed your concerns, we would be grateful if you could let us know.
>
>
> Thank you so much, and we wish you all the very best!
>
> Sincerely,
>
> Authors

---

### Official Review · Reviewer_w3hk · 2024-11-02

**Soundness:** 2
**Presentation:** 3
**Contribution:** 2
**Rating:** 5
**Confidence:** 4

**Summary:**

This paper examines "discretization mismatch errors" in neural operators, like the Fourier Neural Operator (FNO), which learn mappings between different types of data. The authors find that training these models at low resolutions and applying them at high resolutions can lead to performance issues. They propose a new Cross-Resolution Operator-learning Pipeline to avoid these errors, improving accuracy in tasks requiring precise predictions across multiple resolutions, such as climate modeling and fluid dynamics.

**Strengths:**

1. The phenomenon of "discretization mismatch errors" is a significant issue in operator learning. This paper proposes the CROP framework to address (reduce) this error.

2. The numerical examples in this paper show that CROP achieves improved performance over classic neural operator architectures, with the Fourier Neural Operator (FNO) used as an example.

**Weaknesses:**

1. Compared to approaches like "physics-informed constraints" and "continuous-discrete equivalence (CDE)," the CROP framework takes a more practice-oriented perspective. That is, the CROP framework is not inherently discretization-invariant. Specifically, it can only operate on fixed scales chosen *a priori*. This raises a question: how can the CROP framework be adapted to handle infinitely fine resolutions? Is there a formulation of CROP that accommodates an infinite range of scales?

2. The theoretical treatment of "discretization mismatch errors," particularly the results in Proposition 4.4, warrants closer examination:
   1. The authors only provide an upper bound for the error $E_{MN}$. Mathematically, this does not necessarily imply that the "discretization mismatch errors" will vary in line with the trends of the upper bound.
   2. Given the upper bounds in Lemma B.2, how should we interpret the conclusion that "discretization mismatch errors" increase as $M$ grows? Interestingly, it appears that the upper bound does not depend on $M$.
   3. The derivation of Lemma B.2 is relatively straightforward, especially for the terms $\omega^L$ and $\prod_{\ell=1}^L C_\ell$, making the conclusion that "discretization mismatch errors" may increase with $L$ and $\omega$ unsurprising. This could likely be generalized to other neural operator models as well.

3. The numerical results in Table 2 could be expanded. Specifically, it would be beneficial to include results from training resolutions of $127 \times 128$ or even $256 \times 256$. Many recent studies on neural operators (such as papers from ICLR 2024) include similar numerical tests, but their results appear more favorable than those shown in Table 2 of this manuscript.

4. I have concerns about the numerical tests in Table 4, as the operators used in the Darcy and Poisson problems are both linear, while the neural operators being tested (FNO, CNO, U-Net, ResNet, DeepONet) are nonlinear. Moreover, the results for the Poisson problem in Table 7 do not demonstrate a significant advantage of CROP over the standard FNO.

**Questions:**

Please refer to the questions in the weaknesses section.

---

> ### Author Response · Authors · 2024-11-15
> **Author Rebuttal to Reviewer w3hk: Part I**
>
> Thank you very much for your valuable feedback! We have revised the manuscript accordingly and provided our responses below.
>
> > W1: Infinitely fine resolution of CROP and comparison with CDE and Physics-informed constraints
>
> 1. **CROP handles inifitely fine resolutions with a fully connected point-wise network, which aims to recover high frequencies lost due to the prior band-limit assumptions**. We provide theoretical support for this (see Remark 4.5), explaining why this network effectively recovers high frequencies.
>
> 2. Moreover, we would like to note that **CDE cannot handle infinitely fine resolutions** nor can it enable the underlying network to perform cross-resolution tasks.
>
> 3. Physics informed constraints (PICs) are difficuilt to train and is oftentimes rectricting. We focus on the data-learning perspective while PICs require explicit PDE forms, which can even be trained without data if we have the full governing PDE. **We do not think there is a fair comparison between our method and PIC.**
>
> 4. Your question is very insightful and coincides with the message we aim to convey in our work, but we would rephrase it as, "Can any existing grid-based operators truly handle infinitely fine resolutions?" A common misconception in the field is that they simply can, as with FNO’s purported ability to manage fine resolutions. **However, our work demonstrates that the answer is, in fact, NO; FNO and other grid-based operators don’t achieve this.** We urge the community to pay more attention to this issue and propose new solutions. In our paper, we have also discussed the limitations of CROP in terms of handling infinite-dimensional learning in Sec. 6 and provide insights into super resolution for SciML in Appendix E.
>
> > W2.1: The upper bound does not necessarily imply that the "discretization mismatch errors" will vary in line with the trends of the upper bound.
>
> 1. **This is why we included Section 5.1.1, which empirically demonstrates that this issue is present in FNOs**. In Fig. 3(a), we clearly show that **discretization errors rise significantly when the testing resolution differs from the training resolution**. Similarly, in Fig. 3(b), we show that **discretization errors increase with the number of layers**. These experiments provide clear evidence for our claims.
>
> 2. Yes, while the results provide an upper bound—meaning the trend doesn't always hold precisely, as evidenced by the high variance in Fig. 3(b)—we want to emphasize that in practical super-resolution tasks, testing a model's super-resolution capability on specific test sets may not even be feasible. Therefore, **improving performance in the worst-case scenario is essential**. Moreover, it can be observed that, although it does not follow precisely, the overall error accumulation trend is clear.
>
> > W2.2: Omitted M in Lemma B.2.
>
>   1. What we have shown in Lemma B.2 is that, if for training at a fixed finer resolution $M$, **the discretization mismatch error increases as the inference resolution $N$ decreases** (i.e., as the testing resolution gets lower, the upper bound of the DMEs gets larger). This works the other way as well.
>
>
>   2. Basically, as long as the inference resolution differs from that of training, it will suffer from DMEs, whether it is super-resolution or sub-resolution (lower resolution), as demonstrated in the experimental results in Sec. 5.1.
>
>   3. We simplified the proof a little for the general audience, and along the way, this caused some confusion. **We have updated the descriptions in Proposition 4.4 and included the full proof without simplification to avoid any confusion.** The proof is similar to that of the previous one, especially in the treament of the terms $\omega^L$ and $\prod_{\ell=1}^L C_{\ell}$, although additional efforts are included on the Fourier analysis.
>
>   4. We have also added a sentence (line 288) to help readers better understand this. Additionally, we have included interpretations of the upper bound after the proof (lines 820 and 874).

---

> > ### Comment · Reviewer_w3hk · 2024-11-20
> >
> > The revised Lemmas B.1 and B.2, along with their updated proofs, look much better. The improvements also make Proposition 4.4 more coherent and solid in conveying the mathematical insights.

---

> > ### Comment · Reviewer_w3hk · 2024-11-20
> > **More comments and concerns**
> >
> > Thank you for the responses. I have some additional comments and concerns that I believe are more critical for this manuscript:
> >
> > 1. In Figure 1, why does the “Difference between the Inputs” appear almost zero? The same input functions with different resolutions are typically not equal to each other. Please clarify this point.
> >
> > 2. I assume that the CROP framework revises an existing neural operator $\mathcal{L}$ to
> >    $$
> >    Q_{\text{CROP}} \circ \mathcal{L} \circ P_{\text{CROP}}
> >    $$
> >    as shown in Figure 2 and Section 4.3. Based on this, I have the following questions:
> >    - (a) What is the setup for the band-limited convolution kernel used in this work for each numerical test?
> >    - (b) As proven and discussed in the CNO paper (2023) and [6], even if the operator $\mathcal{L}$ maps between two band-limited spaces $\mathcal{B}_w$, a specific architecture and activation function are required to preserve discretization invariance. How, then, can the authors claim that arbitrary neural operator architectures can be applied within the CROP framework? It is suggested that the authors test additional architectures for $\mathcal{L}$.
> >    - (c) Is it possible to provide any theoretical justification for why the CROP framework can reduce "discretization mismatch errors," based on the analysis in the appendix and Proposition 4.4?
> >
> > Finally, since the results in Table 4 concern only linear operators, this manuscript essentially addresses the NS problem for reducing "discretization mismatch errors" and the learning capability tests in Sections 5.1 and 5.2. Thus, more numerical tests are strongly recommended in both directions. Examples may include the Darcy problem ($a(x) \to u(x)$), the Burgers' equation, the Helmholtz equation, and others.

---

> ### Author Response · Authors · 2024-11-15
> **Author Rebuttal to Reviewer w3hk: Part II**
>
> > W2.3: Derivation of Lemma B. 2 is relatively straightforward and could likely be generalized to other neural operator models as well.
>
> 1. It’s **subjective** to consider whether this proof is straightforward; while this is not a paper on numerical analysis, **we value the insights it offers into why FNO struggles with zero-shot super-resolution tasks**.
>
> 2. You’re absolutely correct that this could likely extend to other grid-based neural operators; in fact, line 215 (first paragraph under Sec. 4) of our paper states, "the concepts and analysis can be extended to other mesh-based neural operators, such as the Wavelet neural operator (Gupta et al., 2021)."
>
> 3. Our intention isn’t to single out FNO as the only model that falls short of acting as an infinite-dimensional operator; we demonstrate this issue with FNO simply because it is the most well-known and revolutionary framework. Instead, we aim to **highlight an overlooked aspect of discretization mismatch errors that can inspire the community to advance neural operator design**.
>
>
> > W3.1: Addtional experiments on training resolutions $128 \times 128$ and $256 \times 256$
>
> 1. We provide only the results from training resolution of $64 \times 64$ as **an example to demonstate this issue of discretization mismatch errors**.
>
> 2. We provide the results from training resolutions of $128 \times 128$ and $256 \times 256$ here, which again demonstates this issue. **If you find it neccessary to include this in the paper, please let us know; we will include this result in the appendix.**
>
> Training resolution $128 \times 128$ (over $10$ runs):
>
> | Neural Operator        | **$256 \times 256$**       | **$128 \times 128$**       | **$64 \times 64$** | **$32 \times 32$**       |
> |------------------------|----------------------------|-----------------------------|---------------------------------|---------------------------|
> | FNO                    | $1.47 (\pm 0.46)$          | $0.58 (\pm 0.03)$           | $3.19 (\pm 1.03)$               | $11.77 (\pm 3.65)$        |
> | CROP (ours)            | **$0.54 (\pm 0.05)$**      | **$0.54 (\pm 0.05)$**       | **$0.54 (\pm 0.05)$**           | **$0.54 (\pm 0.05)$**    |
>
>
> Training resolution $256 \times 256$ (over $5$ runs as it takes ~16x training time for FNO compared to $s = 64$):
>
> | Neural Operator        | **$256 \times 256$**       | **$128 \times 128$**       | **$64 \times 64$** | **$32 \times 32$**       |
> |------------------------|----------------------------|-----------------------------|---------------------------------|---------------------------|
> | FNO                    | $0.61 (\pm 0.05)$          | $3.41 (\pm 0.57)$           | $7.73 (\pm 1.97)$               | $13.43 (\pm 3.92)$        |
> | CROP (ours)            | **$0.57 (\pm 0.06)$**      | **$0.57 (\pm 0.06)$**       | **$0.57 (\pm 0.06)$**           | **$0.58 (\pm 0.06)$**    |
>
> > W3.2: Better super-resolution results in other papers
>
> 1. Different settings and datasets might lead to different results; we use the **exact same dataset and setup (including hyperparameters) from the original FNO paper [1]**. This is a time-dependent PDE with auto-regressive rollouts, and the discretization mismatch errors accumulate through time (see our Fig. 3 (a)).
>
> 2. Moreover, **FNO-based models struggling with super resolution tasks can be found in several recent works, spanning both autoregressive rollout taks and one-to-one mapping tasks.** To name a few:
>
> In [3], FNO has an relative $\ell_2$ error of **$8.41$%** (from their Table 1 row 1) on the Navier Stokes equation with $\nu=1 \mathrm{e}-4$ and an super resolution test error of **$43.02$%** (from their Table 2 row 1).
>
> In [4], an FNO trained on a resolution of $43\times 43 (s=43)$ reported the following relative $\ell_2$ errors (from their Table 3) on Darcy flow equations:
>
> | Resolution (s)      | Reported Error |
> |--------------------|----------------|
> | 61 x 61 (s=61)     | 0.1164         |
> | 85 x 85 (s=85)     | 0.1797         |
> | 141 x 141 (s=141)  | 0.2679         |
> | 211 x 211 (s=211)  | 0.3160         |
> | 421 x 421 (s=421)  | 0.3631         |
>
> In [5], an FNO reported the following relative mean square errors (from their Table 5) on Darcy flow equations:
>
> | Resolution       | Reported Error           |
> |------------------|-------------------|
> | $ \frac{1}{2} \times $ | $ 1.475 \cdot 10^{-1} $ |
> | $ 1 \times $           | $ 5.867 \cdot 10^{-2} $ |
> | $ 2 \times $           | $ 8.646 \cdot 10^{-2} $ |
> | $ 4 \times $           | $ 7.731 \cdot 10^{-2} $ |
>
> In [5], an SFNO reported the following relative mean square errors (from their Table 5) on Spherical Shallow Water Equations:
>
> | Resolution       | Reported Error                      |
> |------------------|----------------------------|
> | $ \frac{1}{2} \times $ | $ 1.342 \cdot 10^{-3} $ |
> | $ 1 \times $           | $ 9.220 \cdot 10^{-4} $ |
> | $ 2 \times $           | $ 3.830 \cdot 10^{-3} $ |
> | $ 4 \times $           | $ 4.419 \cdot 10^{-3} $ |

---

> ### Author Response · Authors · 2024-11-15
> **Author Rebuttal to Reviewer w3hk: Part III**
>
> > W4: Concerns about the numerical tests on the Darcy flow and the Poisson problemS
>
> 1. These two PDEs are commonly used in many studies. For example, Darcy flow is tested in [2, 4, 5], and the Poisson equation is tested in [6, 7, 8]. Moreover, Darcy flow is a linear PDE; however, the operator itself is not linear (see [2], page 2, last sentence of the paragraph containing equation (2) of the ICLR 2020 version).
>
> 2. The results in Table 7 do not demonstrate a significant advantage, further underscoring our concerns about zero-shot super-resolution tests discussed in line 1147. We could have only provided example in which we demonstrate significant advanges; however, we would like to be upfront and to motivate the community for further inverstigation and innovations. Additionally, in Appendix E, we provide more discussion on our vision and aspirations for the study and development of methods with insights from mathematics and physics for super-resolution tasks in the SciML field.
>
> References:
>
> [1] Fourier Neural Operator for Parametric Partial Differential Equations, Zongyi Li et al., ICLR 2021
>
> [2] Neural Operator: Graph Kernel Network for Partial Differential Equations, Zongyi Li et al., ICLR 2020
>
> [3] Group Equivariant Fourier Neural Operators for Partial Differential Equations, Jacob Helwig et al., ICML 2023
>
> [4] Improved Operator Learning by Orthogonal Attention, Zipeng Xiao et al., ICLR 2024
>
> [5] Neural Operators with Localized Integral and Differential Kernels, Miguel Liu-Schiaffini el al., ICML 2024
>
> [6] Convolutional Neural Operators for robust and accurate learning of PDEs,
> Bogdan Raonić, et al., NeurIPS, 2023
>
> [7] BENO: Boundary-embedded Neural Operators for Elliptic PDEs, Haixin Wang et al., ICLR 2024
>
> [8] Learning the boundary-to-domain mapping using Lifting Product Fourier Neural Operators for partial differential equations, Aditya Kashi et al., ICML 2024

---

> > ### Author Response · Authors · 2024-11-17
> > **Friendly Reminder: Rebuttal Period for ICLR Submission**
> >
> > Dear Reviewer w3hk,
> >
> > We are writing to kindly remind you that **we posted our response almost 3 days ago, if you have any additional feedback, concerns, or questions regarding our response, we would greatly appreciate hearing from you.**
> >
> > We have made significant efforts to address your concerns. Specifically, we refined the proof and added more intuitive explanations and interpretations of the theoretical statement. Additionally, we conducted the suggested experiments to further underscore our findings on DMEs. Please feel free to let us know if any further clarification or elaboration is required.
> >
> > Thank you for your time and effort in reviewing our paper and engaging in the discussion process.
> >
> > Best Regards
> >
> > Authors

---

> > ### Comment · Reviewer_w3hk · 2024-11-20
> > **W4: Concerns about the numerical tests on the Darcy flow and the Poisson problems**
> >
> > "Moreover, Darcy flow is a linear PDE; however, the operator itself is not linear (see [2], page 2, last sentence of the paragraph containing equation (2) of the ICLR 2020 version)."
> >
> > Please note that the operator mentioned in [2] is from $a(x) \to u(x)$ in Darcy problem which is of course indeed a nonlinear operator. However, the problems in both Poisson and Darcy in this manuscript are from $f(x) \to u(x)$ which are exactly both linear operators.

---

> ### Author Response · Authors · 2024-11-22
> **Second Round Author Rebuttal to Reviewer w3hk: Part I**
>
> We sincerely thank the reviewers for their insightful comments and valuable suggestions. Overall, we have conducted more experiments and have further revised our paper; we will provide the responses below.
>
> > The revised proof is better and conveying the mathematical insights.
>
> - **Thank you for your comments. We are glad that you are satisfied with the updated proof.**
>
> > Darcy Flow equation
>
> - You are absolutely correct. This is an oversight during the rebuttal, because originally we spent a lot of time doing testing on the Darcy flow from FNO [1] when working on the initial version of this manuscript. The operator for Darcy flow in FNO [1] is non-linear while ours is not. *We do not have any incorrect/plausible statement on this in the manuscript.*
>
> - **We have additionally presented results on the Darcy flow task directly from FNO [1], i.e. the non-linear operator setting of the Darcy flow equation.** (you raised a similar question in your comments, so we provide detailed responses later at the end)
>
> > In Figure 1, why does the “Difference between the Inputs” appear almost zero? The same input functions with different resolutions are typically not equal to each other. Please clarify this point.
>
>   - In line 242, we state "For a discretization of size $N$, we will view it as a function reconstructed by the $N$ Fourier coefficients", and this is consistent with our proof.
>
>   - In this view, the difference between two inputs of varying discretizations can appear almost zero because the low-frequency components are very close, and the high-frequency components of the higher-resolution input have negligible amplitudes. Additionally, the color-bar scale (in Figure 1 of our paper) is based on the output difference map, which means that the discrepancies in the inputs are much smaller than those in the outputs.
>
>
> - **The difference shown in Figure 1 is based on the *real results* from plotting the Navier-Stokes $ 1 \times 10^{-3}$ example from FNO [1] data and code**, where input resolutions are $64 \times 64$ and $256 \times 256$, respectively, while the model was trained on $64 \times 64$. The discrepancy between these two input discretizations is significantly much smaller compared to the differences observed in their corresponding outputs.
>
> ### Questions on CROP
> > (a) Setup for the band-limited convolution kernel
>
> - **This convolution kernel is realized using a limited number of (learnable) Fourier coefficients representing bandlimited convolution kernel functions.** In other words, it is parameterized directly by complex parameters. However, **two key factors** in CROP layers are required to maintain bandlimits.
>
>    - First, in line 302, we state: "As we consider a band-limited function space, we can fix the discretization of the latent feature function in accordance with the band-limit we choose." For example, if the band-limit of the convolution kernels is $K$, the intermediate discretization will be $2K$. We apply activation functions only after fixing the discretization. **This removes any DMEs, as the intermediate discretizations are always consistent with the predefined band-limit in the convolution kernels.**
>
>   -  Second, we do not have residual connections from the input to the output of the CROP lifting layer directly, as this might break the predefined band-limit since the input is not in the band-limited space. There are two possible choices: one is to avoid using residuals for CROP lifting layers (still have residual connections in other layers), and the other is to build residual connections using the Fourier projection of the input into the band-limited function space. The second approach worked better for the examples we tested, so we used the second approach for all experiments.
>
> - In all numerical tests, the band-limit is chosen to correspond with the truncation mode in FNO for a **fair comparison**.

---

> ### Author Response · Authors · 2024-11-22
> **Second Round Author Rebuttal to Reviewer w3hk: Part II**
>
> > (b) Specific architecture and activation function are required and different achitectures for intermidiate neural operator.
>
> - **The perspectives of CNO [2] and our work are different. Their specific architecture and activation functions are designed to ensure continuous-discrete equivalence (CDE), not cross-resolution abilities.** They analyze several existing operators and determine whether or not they exhibit CDE properties (if that's what is meant by "specific architecture"). For some operators, specific activation functions are not necessary to ensure CDE. For example, DeepONet is a CDE operator, without specific choices of activation functions, assuming uniform-grid outputs and a fixed band-limited target function space (see Appendix B.3 in [3]).
>
> - **Our target is to study DMEs, find the root cause, and reduce DMEs to enable cross-resolution inferences or even training, which is fundamentally different from the perspectives and aims of CNO/CDE.** These differences in focus are stated in Sec. 2 "Relations with Prior Work". CDE does not imply the ability to perform cross-resolution tasks (downsampling and interpolation are required for cross-resolution, as described in Appendix C.4 of [2]). Additionally, this is noted in [4] (see their Table 1), and they state: "they cannot be applied to different resolutions without relying on explicit up-/downsampling which may degrade performance, e.g., CNO." In fact, their main innovation, the up-and-down activations applied to U-net, can be applied to any grid-based operator. **We show in Table 2 of our manuscript that even FNO with such conventions still suffers from DMEs.**
>
> - **In the table below, we have included additional results for CNO under the same settings as in Table 2 (cross-resolution results) in our paper**, further demonstrating the statement made in [4] and the inefficiency of CNO in performing cross-resolution tasks. **We have also included these results in Table 2 in the revision of our paper.**
>
>
> | Neural Operator        | **$256 \times 256$**       | **$128 \times 128$**       | **$64 \times 64$** | **$32 \times 32$**       |
> |------------------------|----------------------------|-----------------------------|---------------------------------|---------------------------|
> | FNO                    | $4.65 (\pm 1.47)$          | $3.35 (\pm 0.97)$           | $0.58 (\pm 0.04)$               | $9.45 (\pm 2.92)$        |
> | CNO (without interpolation and down-sampling)                    | $73.19(\pm 3.77)$          | $58.63(\pm 3.30)$           |$ 1.96 (\pm 0.13)$              |  $77.23(\pm 7.97)$        |
> | CNO (with bicubic interpolation)                    | $4.17 (\pm 1.46)$          | $2.95 (\pm 1.06)$           | $ 1.98 (\pm 0.14)$               | $7.69(\pm 1.69)$        |
> | CROP (ours)            | **$0.54 (\pm 0.05)$**      | **$0.54 (\pm 0.05)$**       | **$0.54 (\pm 0.05)$**           | **$0.54 (\pm 0.05)$**    |
>
> - Cross-resolution is achieved by up/down sampling and interpolation for CNO. Training resolution $64 \times 64$ (by the way, CNO is very slow, ~14 times slower than FNO on Nvidia RTX A6000, to train under this Navier Stokes autoregressive setting).
>
> > (c) CROP framework can reduce "discretization mismatch errors"
>
> - **Yes**, theoretical justifications are possible and straightforward. We provide the high-level ideas here.
>
>   - As stated above in our response to (a), the CROP lifting layer applies activations only after fixing the discretization. That is to say, $N = M$ after this step in Lemma B.2. Thus, the intermediate layers definitely do not suffer from DMEs.
>
>   - For the lifting layer, it does not amplify DMEs as activations are applied only after fixing the discretization (the $\omega$ term does not affect this layer).
>
>   - For the projection layer, the goal is to recover some high frequencies lost due to bandlimiting. As such, DMEs in this layer are not desired, because minimizing DMEs would result in all the high frequencies being $0$, similar to lower-resolution inferences.

---

> ### Author Response · Authors · 2024-11-22
> **Second Round Author Rebuttal to Reviewer w3hk: Part III**
>
> > Additional Numerical Tests on more non-linear PDEs
>
> - **Our main objective is to test the abilities of CROP to perform multi-spatio-learning tasks. All the examples we include demonstrate local changes, aligning with our objective.** We understand your concerns about the linearity of these PDEs. Therefore, we have included an additional non-linear example.
>
>    - We include the exact same Darcy flow example from FNO [1] (Exactly the same data as the authors of [1] provided). **The results are updated in Table 4 in our revision. We also included cross-resolution performance in Table 7 in Appendix D.2.** We also provide it here for your convenience.
>
> (Table 4) Comparison with baselines (same resolution inference):
>
> |                | **Nonlinear Darcy**      |
> |----------------|--------------------------|
> | **FNO**        | $0.68\pm 0.03$        |
> | **CNO**        | $0.59\pm 0.04$        |
> | **U-Net**      | $1.34\pm 0.03$        |
> | **ResNet**     | $7.26\pm 0.59$        |
> | **DeepONet**   | $2.99\pm 0.10$        |
> | **CROP (ours)**| **$0.51\pm 0.01$**    |
>
> Clearly, CROP outperforms all baselines.
>
> (Table 7) Cross-resolution test (Training resolution is $85 \times 85$.):
>
> |                | **421 × 421**         | **141 × 141**         | **85 × 85**                 | **43 × 43**         |
> |-------------------------|-----------------------------|-----------------------------|-----------------------------|---------------------|
> | **FNO**        | $0.87 \pm 0.04$       | $0.75 \pm 0.03$       | $0.68 \pm 0.03$             | $0.97 \pm 0.03$     |
> | **CROP (ours)**| **$0.52 \pm 0.01$**  | **$0.51 \pm 0.01$**  | **$0.51 \pm 0.01$**        | **$0.57 \pm 0.01$**|
>
> Clearly, CROP is more consistent for cross-resolution tasks while FNO suffers from DMEs.
>
>  - Note that for this particular example, we used the **exact same data** as in the FNO paper. **In the original paper, they reported an error of $1.08$% (Table 4 in FNO paper); later on, the FNO authors updated their architecture, and we use their *updated architecture* and obtained an error of $0.68$% (which is better than $1.08$%). If we were to use the original reported error or trained model, worse performance would be reported for FNO.** Similarly, for CNO, we used their architecture directly and performed model selection the authors provided in their git repo. Using their original released architecture, which yielded the best performance during model selection, we obtained an error of $0.59$%. For other models, we used implementations directly from their respective works or popular implementations, as detailed in Appendix D.5. Additionally, refer to Table 1 from [5]; they conducted experiments on exactly the same data from FNO [1]. For your convenience, we summarize the results in Table 1 from [5]:
>
>     - They report $1.08$% for FNO, which is from the original paper. We report much better results for the FNO baseline. **Additionally, the lowest error achieved in [5] on this dataset is $0.69$%, whereas we reported better results than this for both CNO ($0.59$%; they did not compare with CNO in their paper) and CROP ($0.51$%).**
>
> References
>
> [1] Fourier Neural Operator for Parametric Partial Differential Equations, Zongyi Li et al., ICLR 2021
>
> [2] Convolutional Neural Operators for robust and accurate learning of PDEs,
> Bogdan Raonić, et al.
>
> [3] Representation Equivalent Neural Operators: a Framework for Alias-free Operator Learning, Francesca Bartolucci et al., NeurIPS 2023
>
> [4] Neural Operators with Localized Integral and Differential Kernels, Miguel Liu-Schiaffini el al., ICML 2024
>
> [5] Improved Operator Learning by Orthogonal Attention, Zipeng Xiao et al.,  ICML 2024

---

> ### Author Response · Authors · 2024-11-25
> **Discussion Period Ending Soon**
>
> Dear Reviewer w3hk,
>
> Thank you for your thoughtful feedback and constructive suggestions on our paper throughout the review and rebuttal period. Based on your comments, we have made significant efforts to address the concerns raised. We believe these updates have substantially improved our paper.
>
> As the discussion period ends on **Nov 26, 2024 (Anywhere on Earth)**, we kindly await your feedback or reassessment of our paper. Please let us know if your concerns have been addressed; if yes, we kindly request you update your initial ratings. We are happy to answer any further questions you have.
>
> Thank you once again, and we wish you all the very best :)
>
> Sincerely,
>
> Authors

---

> > ### Comment · Reviewer_w3hk · 2024-11-25
> >
> > Thank you for the responses and updates provided in the second round. I believe the quality of the manuscript has improved, and several obvious mistakes have now been addressed. So, I raised my score. However, I still have the following two main concerns:
> >
> > From my perspective, the final CROP framework does not appear particularly novel. The proposed framework seems almost too straightforward, as it primarily introduces a lifting and projection operator, which have already been included in standard FNO and others (but not for this purpose specifically). Additionally, I strongly recommend conducting more systematic numerical tests to evaluate both the discrete mass error (DME) and overall accuracy. Suggested examples include the Darcy problem with rough coefficients,  the Burgers' equation, the Helmholtz equation, among others.
> >
> > While I have some reservations about the scientific contribution from a computational perspective, I am happy to leave the final decision to the Area Chair (AC). This reflects my personal view that the scientific impact may not be particularly strong within the domain of scientific computing. However, it might be of interest to the general machine learning community.

---

> ### Author Response · Authors · 2024-11-26
> **Third Round Author Rebuttal to Reviewer w3hk: Part I**
>
> Dear Reviewer w3hk,
>
> Thank you for taking the time to review our updates and reassess our work. We really appreciate your hard work! We provide our responses to your remaining concerns below. We've also revised the paper to include more discussions.
>
> > CROP layers
>
> 1. **CROP is fundamentally different from the lifting and projection layers in FNO.** While we have discussed this difference in the paper and in previous responses during rebuttal, we provide further clarification as follows:
>
>    - First, the **FNO maintains the same discretization throughout the process**, starting from the lifting layer, continuing through every intermediate layer and the projection layer. In contrast, CROP focuses on mapping functions into band-limited function spaces; **in CROP, the discretization determined by the band-limits may differ from that of the input.**
>
>    - Second, **the lifting layer in FNO only transforms the input channels through a linear operation along the channel dimension**, where each channel is merely a linear combination of the input channels. **FNO lifting lacks the ability to learn global patterns**. In contrast, **CROP employs global (band-limited) convolution kernels to lift the input functions. This enables CROP to capture global patterns during the lifting process**, providing the intermediate neural operator with richer and more comprehensive information for further processing.
>
>     - Similarly, **the CROP projection layer can operate on different discretizations**. It reconstructs the lost high-frequency information using the band-limited feature functions produced by the intermediate neural operator.
>
> > Recommended Experiments
>
> 1. **We have included most widely-used datasets from the machine learning community**, such as the ones from the FNO and CNO papers. These examples are accompanied by datasets or data generation scripts that are widely recognized and extensively used. Also, **we already have a 1D Burgers’ equation example in the paper (Appendix C.1)**.
>
> 2. Regarding the rough Darcy and Helmholtz examples, we can hardly find reliable and publicly available resources for these two datasets. So we are striving to generate these data by ourselves; however, we are experiencing difficulties in writing stable solvers and generating meaningful datasets within a short time frame.
>
>    - We could only find these two testing examples in [1, 2, 3], but the Git repository link in [1] (Sec. 3.9) is not working, and we couldn’t find open-source code or data from [2].
>
>    - We do have access to the dataset generated in [3]. However, the data is provided only at a $101 \times 101$ resolution; a higher resolution dataset is preferred to study the DMEs.
>
>    We are now working on generating the data ourselves, but given the time constraints, it will be extremely difficult to produce results before the revision deadline (ICLR only allows to revise papers until Nov. 27th). We can continue to post responses on Openreview. Do you have recommendations on where we can obtain high-quality and high-resolution data for our purposes? If you happen to know some, we would greatly appreciate it if you can let us know. After we obtain some results, we will post them on Openreview and include the results in the next version of our paper if you suggest so.
>
> 3. **We believe that the experiments in our paper are sufficient to demonstrate that neural operators suffer from DMEs and that CROP can mitigate this issue (Figures 3, 4, 5, and Tables 2, 5, 7). Additionally, experiments on well-established datasets demonstrate that the learning capabilities of our approach surpass or are at least on par with the baselines (Figure 5 and Tables 2, 3, 4, 7).** We greatly value your insights into including more experiments on the challenging problems you mentioned. **We have referred to these three papers and included a discussion in Sec. 6 of our revision. We have some discussions that it would be valuable to conduct further studies on additional challenging learning tasks inspired by these works.**

---

> ### Author Response · Authors · 2024-11-26
> **Third Round Author Rebuttal to Reviewer w3hk: Part II**
>
> > Computational perspective and impact in the domain of scientific computing
>
> 1. **We believe that our findings are significant for the field of operator learning, a class of ML surrogate solvers.** We are the first to identify DMEs (Sec. 4.1), provide theoretical analyses (Sec. 4.2), verify theoretical conclusions with empirical experiments (Sec. 5.1), and propose a solution (Sec. 4.3). We believe our findings will motivate future research to explore this issue further and develop improved designs for neural operators.
>
> 2. We deeply appreciate your perspectives on traditional scientific computing, as we believe that computational perspectives and ideas can advance the edges of AI methods in providing fast surrogates for physical simulations. This is precisely why, in Appendix E, we include additional discussions on super-resolution for SciML tasks. We believe that the insights and knowledge of applied mathematicians and computational physicists are invaluable, because this will help our work convey impacts in the broad scientific computing communities (not just the ML communities).
>
> References
>
> [1] Mitigating spectral bias for the multiscale operator learning, Xinliang Liu et al., Journal of Computational Physics 2024
>
> [2] Dilated convolution neural operator for multiscale partial differential equations, Bo Xu et al.
>
> [3] The Cost-Accuracy Trade-Off In Operator Learning With Neural Networks, Maarten de Hoop el al., Journal of Machine Learning 2022

---

> ### Author Response · Authors · 2024-12-01
> **Third Round Author Rebuttal to Reviewer w3hk: Part III**
>
> Dear Reviewer WGpV,
>
> Thank you for your engagement during the review and rebuttal period. We have been working diligently to deliver the numerical results as you suggested.
>
> 1. **We have obtained results on the Helmholtz equation [1, 2, 3].** We learn the nonlinear mapping from the wavespeed field to the solution function.
>
>    - The dataset is directly from [3] in the Caltech database of resolution $101\times 101$. We use the first $800$ samples as the training set and samples $801-1,000$ as the test set. The data is normalized to [0, 1] based on the training data. Other implementation details follow those in our paper.
>
>
> 2. We present the results in the Table below for your convenience. **Clearly, CROP outperforms all baseline models due to its multi-spatial-learning capabilities.** As we can not revise our paper now, if you think it necessary, we will include these results in the next revision.
>
>
> (Table) Comparison with baselines (same resolution inference):
>
> |                | **Helmholtz**      |
> |----------------|--------------------------|
> | **FNO**        | $0.68\pm 0.01$        |
> | **CNO**        | $0.87\pm 0.02$        |
> | **U-Net**      | $1.51\pm 0.06$        |
> | **ResNet**     | $14.86\pm 0.53$        |
> | **DeepONet**   | $3.03\pm 0.07$        |
> | **CROP (ours)**| **$0.52\pm 0.02$**    |
>
> 3. There is no high-resolution data provided for this dataset. While we are attempting to generate high-resolution data, it is extremely difficult to do so within a short timeframe. We will add this discussion in the next version along with the table above.
>
>    - Discretizing the Helmholtz equation may lead to *ill-conditioned* systems when the wave number is high.
>    - High-order finite element methods, such as high-order discontinuous Galerkin, are required to faithfully solve such a challenging system. Developing such solvers and generating a large amount of data is a complex and time-consuming task.
>
>
> 4. **As mentioned in the previous reply, we believe we have sufficient numerical tests to demonstrate our main purposes in the paper.**
>
> We look forward to hearing back from you. Your feedback is essential to improving our work, and we greatly appreciate your insights. Please let us know if you have any further questions or suggestions.
>
>
> [1] Mitigating spectral bias for the multiscale operator learning, Xinliang Liu et al., Journal of Computational Physics 2024
>
> [2] Dilated convolution neural operator for multiscale partial differential equations, Bo Xu et al.
>
> [3] The Cost-Accuracy Trade-Off In Operator Learning With Neural Networks, Maarten de Hoop el al., Journal of Machine Learning 2022

---

### Meta-Review · Area_Chair_pSWm · 2024-12-22

**Metareview:**

This paper studies the discretization mismatch errors that can arise in neural operators. Specifically, it is shown that these errors can degrade the performance of Fourier Neural Operators (FNOs) when the input and output data differ in resolution. To address this issue, the authors propose a Cross-Resolution Operator-learning Pipeline (CROP) and provide both a theoretical foundation and strong empirical results.

These insights are relevant to the neural operator community. By advancing our understanding of discretization mismatch errors, this work can inspire further research and contribute to the development of more robust neural operators for scientific computing. Reviewers commend the soundness of the proposed model and analysis, as well as the compelling empirical findings that illuminate the limitations of FNOs in cross-resolution settings. Furthermore, the paper is clearly written, making the authors’ arguments easy to follow.

Although reviewers noted several weaknesses, the authors effectively addressed these concerns during the rebuttal phase. Overall, I feel that the strengths outweigh the weaknesses. Thus, I recommend accepting this submission.

**Additional Comments On Reviewer Discussion:**

The authors effectively used the rebuttal period to address all major concerns. They also provided a range of additional experiments, and carefully revised the manuscript

---

### Decision · Program_Chairs · 2025-01-22

Accept (Poster)